**The evolution of aerosols mixing state derived from a field campaign in Beijing: implications to the particles aging time scale in urban atmosphere**

**Jieyao Liu[1], Fang Zhang[2], Jingye Ren[3], Lu Chen[4], Anran Zhang[2], Zhe Wang[2], Songjian Zou[2], Honghao Xu[2], Xingyan Yue[1]**

[1] School of Geographical Sciences, Hebei Normal University, Shijiazhuang, China

[2] School of Civil and Environmental Engineering, Harbin Institute of Technology (Shenzhen), Shenzhen, China

[3] Xi'an Institute for Innovative Earth Environment Research, Xi'an, China

[4] Faculty of Geographical Science, Beijing Normal University, Beijing, China

Correspondence to: Fang Zhang (zhangfang2021@hit.edu.cn)

**Abstract**

The mixing states and aging time scale of aerosol particles play a vital role in evaluating their climate effects. Here, by using the field measurement at a site of the urban Beijing, we have identified four different real-time mixing patterns of size-resolved particles, which are defined as less hygroscopic (LH) internally-mixed, externally-mixed, transitional externally-mixed and more hygroscopic (MH) internally-mixed particles, with atmospheric fraction of 0-10%, 20-46%, 17-24% and 27-56% respectively. The fraction depends on particles size, with the maximum fraction of MH internally-mixed particles at 80 and 110 nm, and the minimum fraction of LH internally-mixed particles across all sizes, implying rapid mixing and aging of ambient particles during the observational period. The diurnal variations of the mixing states of particles in all sizes investigated (40, 80, 110, 150 and 200 nm) present an apparent aging process from externally-mixed to MH internally-mixed, which typically spans

a duration of approximately 5–10 hours from 8:00–10:00 to 15:00–17:00, revealing the
mixing (aging) timescale of aerosols in polluted urban atmosphere. Additionally, our results
suggest that those fine aerosol particles experience aging through both the photochemical
process and non-photochemical growth during the campaign. Furthermore, through a
comprehensive review of the mixing/aging timescale of particles adopted in current models
and derived from observations, we show the great discrepancy between observations and
models, highlighting the importance to parameterize their aging time scale based on more
field campaigns.

**1 Introduction**
The mixing state of atmospheric aerosol particles can affect the hygroscopicity and the
ability to serve as cloud condensation nuclei (CCN), and thus the air quality and climate
(Müller et al., 2017; Xu et al., 2021; Yao et al., 2022; Ge et al., 2024). It has been shown that
the aerosol mixing state is closely related to the hygroscopicity (Chen et al., 2022; Fan et al.,
2020). Ren et al. (2018) predicted the concentration of CCN using five different mixing state
schemes and found that the influence of aerosol mixing state on its activation characteristics
ranged from –34% to +16%. Neglecting particle mixing structure can also lead to significant
overestimation of the aerosol absorption efficiency (Yao et al., 2022). Therefore, it is
important to account for the information of mixing state of ambient particles in climate
models so as to reduce the uncertainty in evaluating their environmental and climate effects.
The mixing states of ambient particles are complex. Particles in areas affected by
primary emissions are mainly in an external mixing state (i.e., the chemical components of
particles exist independently), while aerosols in relatively clean areas are mainly transported
from elsewhere and have a higher degree of internal mixing (Swietlicki et al., 2008; Enroth et

al., 2018; Chen et al., 2022). Internal mixing typically includes uniform composition or core–shell structures (Jacobson et al., 2001). The former refers to the same proportion of species in any part of the aerosol component, while the latter is defined as the mixing state formed by certain chemical components coating or condensing on the surface of other components during the aging process. Also, the mixing state of aerosol particles is variable. Freshly emitted particles undergo various processes, including photochemical and aqueous-phase processes, as well as physical processes such as coagulation and condensation, leading to an increase in their degree of internal mixing. This gradual transition from external to internal mixing characterizes the aging process of particles. The aging timescale varies greatly between clean and polluted areas (Peng et al., 2016; Chen et al., 2017; Ghosh et al., 2021). However, the timescale of aging process of particles was commonly fixed in many models and did not depend on environmental conditions (Chen et al., 2017; Ghosh et al., 2021), which may introduce great uncertainty in the prediction of regional aerosol concentration and the evaluation of aerosol climate effects (Ghosh et al., 2021). Therefore, capturing the temporal scales of the evolution of the aerosol mixing state based on field campaigns is crucial for accurately parameterizing the aging timescale of aerosol particles in models, thereby enhancing the precision of simulations pertaining to the environmental and climatic impacts of aerosols.

At present, some studies have characterized the mixing state and aging process of black carbon (BC) aerosols using different instruments. Transmission electron microscopy (TEM) has been used to determine the mixing state of individual particles in China (Li et al., 2016; Zhang et al., 2023). However, based on TEM technique, a large number of aerosol samples

are required so as to make the results statistically significant, which means a high cost both
on labors and materials. The most important issue is that the mixing state of particles may
change during collection and transportation. Recently, aerosol time-of-flight mass
spectrometry (ATOFMAS) and soot-particle aerosol mass spectrometry (SP-AMS) have been
used to measure the mixing state of BC and coated aerosol species in real time (Liu et al.,
2019; Xie et al., 2020). Overall, previous studies focused more on the mixing state of BC of
single-particles (Saha et al., 2018; Xie et al., 2020; Chen et al., 2020). However, the mixing
state of particles across a population in ambient atmosphere is more complex (Riemer et al.,
2019). According to Winkler's definition, the "internal mixing" means when all particles
within the bulk aerosol populations have the same compositions. The "external mixing"
means when all particles consist of pure species and have distinct compositions (Winkler et
al., 1973; Riemer et al., 2019). Note that in Riemer et al. (2019), the definition refers to the
bulk not the sized-resolved mixing state of aerosol particles. In this study, the size resolved
mixing state of aerosol particles has been identified according to the patterns of the
probability density function of hygroscopic growth factor/hygroscopic parameter ($\kappa$) (Gf-
PDF; $\kappa$-PDF) measured using a humidity tandem differential mobility analyzer (H-TDMA)
(Hong et al., 2018; Shi et al., 2022; Spitieri et al., 2023). For example, the internally-mixed
aerosol populations with diameter of 40 nm are characterized by the single hygroscopic mode
of Gf-PDF/$\kappa$-PDF, while the externally-mixed particles have bi- or trimodal distributions of
Gf-PDF/$\kappa$-PDF. However, most studies based on the H-TDMA measurements only made
qualitative descriptions of the particle mixing state when explaining the variations in the
aerosol hygroscopicity (Wang et al., 2019; Chen et al., 2022; Shi et al., 2022).
In this study, with the aim of obtaining insights into the mixing state and mixing (aging)
time scale of ambient particles in urban area, we have identified four different types of size-
resolved particles mixing states, and characterized their real-time variations using the field
measured hygroscopic growth factor by the H-TDMA in urban Beijing. The evolution of
mixing state of particles with specific sizes was explored to imply the mixing (aging)
timescale of their aging from the diurnal variations during clear and cloudy days. Finally, we
compared the mixing/aging timescale of aerosol particles with that adopted in current models
and other field observations reported in previous literatures.
**2 Methods**
The campaign was conducted at the meteorological tower branch of the Institute of
Atmospheric Physics (IAP), Chinese Academy of Sciences to measure the physical and
chemical properties of particles and the meteorological conditions from 19 May to 18 June
2017 in Beijing. The instruments used in this study were deployed in a container at ground
(the sampling inlet is located at ∼8 m on the meteorological tower). The hygroscopicity of
aerosols with different dry sizes (40, 80, 110, 150 and 200 nm) and particle number
distribution in the size range of 10–550 nm was measured using a H-TDMA and a Scanning
Mobility Particle Sizer (SMPS), respectively. The mass concentration of the non-refractory
chemical compositions in particulate matter with diameter < 1 μm ($PM_1$) was measured by an
Aerodyne high-resolution time-of-flight aerosol mass spectrometer (HR-ToF-AMS) (Xu et al.,
2019), and the BC was measured by a 7-wavelength aethalometer (AE33, Magee Scientific
Corp). More details about the sites, instruments and calibrations can be found in previous
literatures (Zhang et al., 2017; Wang et al., 2019; Fan et al., 2020; Chen et al., 2022). Since
the campaign site was located at the urban area of Beijing, where the atmospheric fine
aerosols are more frequently affected by local traffic and cooking sources (Sun et al., 2015).
In addition, during the observation period of summertime, the site was predominately
affected by the warm and humid southeastern air parcels (Fig. S1); while the dust events
normally originate from northwestern regions occurring in springtime (Li et al., 2020) and
typically with particle sizes in coarse mode (Wang et al., 2023). Therefore, the contribution
from dust aerosols could be negligible in this study.

Here, we mainly describe the interpretation criteria for the mixing state of particles. The

Gf-PDF was obtained by TDMA$_{inv}$ algorithm in this study (Gysel et al., 2009). Then, the $\kappa$-
PDF of size-resolved particles was retrieved according to the $\kappa$-Köhler theory (Petters and
Kreidenweis, 2007). Subsequently, we accurately defined four mixing states based on the
$\kappa$-PDF patterns and the number of peaks in $\kappa$-PDF (Fig. 1, Fig. S2). The $\kappa$-PDF of type 1
exhibits bimodal distributions with a much higher proportion ($> 70\%$) of hydrophobic mode
or less hygroscopic mode (LH; the peak of $\kappa$-PDF occurs at $\kappa < 0.1$), was thus defined as LH
internally-mixed state (type 1). Mixing type 2 (Externally-mixed) is characterized by bimodal
distributions in the $\kappa$-PDF for both LH and more hygroscopic (MH, $\kappa \geq 0.1$) modes, with less
than 30% difference in the fraction of the two modes. The $\kappa$-PDF pattern with trimodal
distributions was named as transitional externally-mixed state (type 3). The $\kappa$-PDF of type 4,
defined as the MH internally-mixed type, was dominated by the MH mode with a fraction
larger than 70%.It should be noted that the four categories defined in our study have covered
all cases of the observational results. In this study, we also calculated the standard deviation
of $\kappa$-PDF ($\sigma$) according to Spitieri et al. (2023), which indicates the degree of dispersion in
the data, and is thought to reflect the mixing degree of particles to a certain extent. Given that
the aerosol particles sampled at the campaign site in this study are rarely affected by other
hygroscopic aerosols (e.g., aerosols from marine sources), the MH internally-mixed particles
(110, 150 and 200 nm) are thus mainly from the aging and growth of smaller particles. In
contrast, the larger LH internally-mixed particles are predominantly from primary emissions.
Therefore, the proportion of MH and LH internally-mixed particles at 100-200 nm can be
used to distinguish the contribution to the internally-mixed fraction of larger particles from
smaller aerosol aging and primary emissions respectively. Specifically, the result shows that a
significant proportion of the internally-mixed particles at larger size originate from the aging
and growth of smaller particles, accounting for 99%, 96% and 92% respectively for 110, 150
and 200 nm particles, whereas the contribution from primary emissions to these larger
internally-mixed particles is less than 10%. It is worth noting that these results may involve
certain uncertainties, as the influence of other sources was not considered. For example, the
more hygroscopic sulfate particles could also be directly emitted from primary sources in
urban atmosphere (Dai et al., 2019); in addition, the aerosols are assumed rarely affected by
the sea salts. Consequently, the contribution of smaller particle aging and growth to the
internally-mixed fraction of larger particles may have been overestimated in this study. More
research works are warranted to further evaluate and quantify such uncertainty in future.

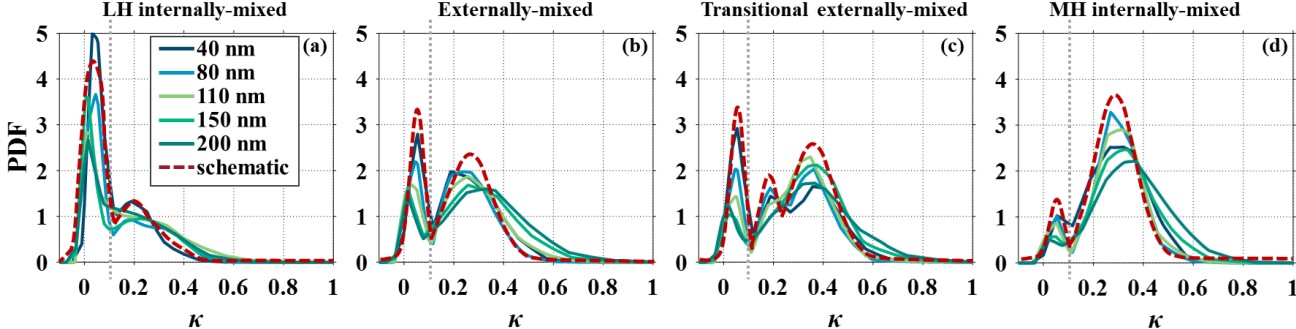


Figure 1. The mean $\kappa$-PDF of five particle sizes and the schematic diagram of $\kappa$-PDF for four
mixing types.
**3 Results and discussion**
**3.1 Overview of the mixing state of size-resolved particles**
Figure 2 shows the time series of size-resolved $\kappa$, and $\kappa$ versus σ. The real-time changes
in the mixing states of particles are represented by different filled colors. The $\kappa$ values vary
from 0 to 0.5 in each of the five particle sizes, accompanied by significant changes in mixing
states. In general, the mixing state of particles gradually transitions from externally-mixed to
MH internally-mixed along with increases in $\kappa$. For example, particles with size of 40 nm are
dominated by LH internally-mixed and externally-mixed at $\kappa < 0.2$; and the transitional
externally-mixed state take up an increasingly proportion often corresponding to $\kappa$ values of >
0.2 and the MH internally-mixed particles occur when $\kappa$ greater than 0.3 (shown by the light
gray dots in Fig. 2).
However, there exists different correlations between $\kappa$ and σ for the five sizes. For
particles with diameter of 40 and 80 nm, the σ increases as the $\kappa$ increases from 0.1 to 0.4
along with slight increase in the fraction of MH internally-mixed particles, showing a positive
correlation. Unlike the small particles, the σ for particles larger than 100 nm (i.e., 110, 150
and 200 nm) exhibits a negative correlation to $\kappa$ variations, showing that the particles were
more MH internally-mixed and with stronger hygroscopicity when the smaller σ values were
derived. The different sources and chemical compositions of aerosol particles could explain
the distinct relationship between σ, mixing states and hygroscopicity among particles of
different sizes. Here, the standard deviation of $\kappa$-PDF, σ, reflects the degree of dispersion in
the pattern of $\kappa$-PDF. Taking 40 nm particles as an example, the obtained larger σ value
indicates more diversity in particles hygroscopicity and chemical compositions, and thus a
higher degree of external mixing state. This is because that, besides the local emissions (Ren
et al., 2023), the 40 nm particles during the campaign were also from the growth of newly
formed particles which are much more hygroscopic (Liu et al., 2021a). Therefore, the
composition of those small particles is more heterogeneous, resulting in both greater σ and $\kappa$.
The cases where both the σ and $\kappa$ of small particles are lower suggest that the particles may
consist of a mixture of different species originating from primary emissions (e.g., BC,
hydrocarbon-like and/or cooking organic aerosols). In contrast, for most of larger particles,
their chemical composition tends to become homogeneous after aging and growth processes,
exhibiting stronger hygroscopicity, a higher degree of internal mixing, and thus the smaller σ.
However, a small number of larger particles with weak hygroscopic properties from primary
emissions coexist with the aged ones in the atmosphere, which would result in larger σ and
higher degree of external mixing.

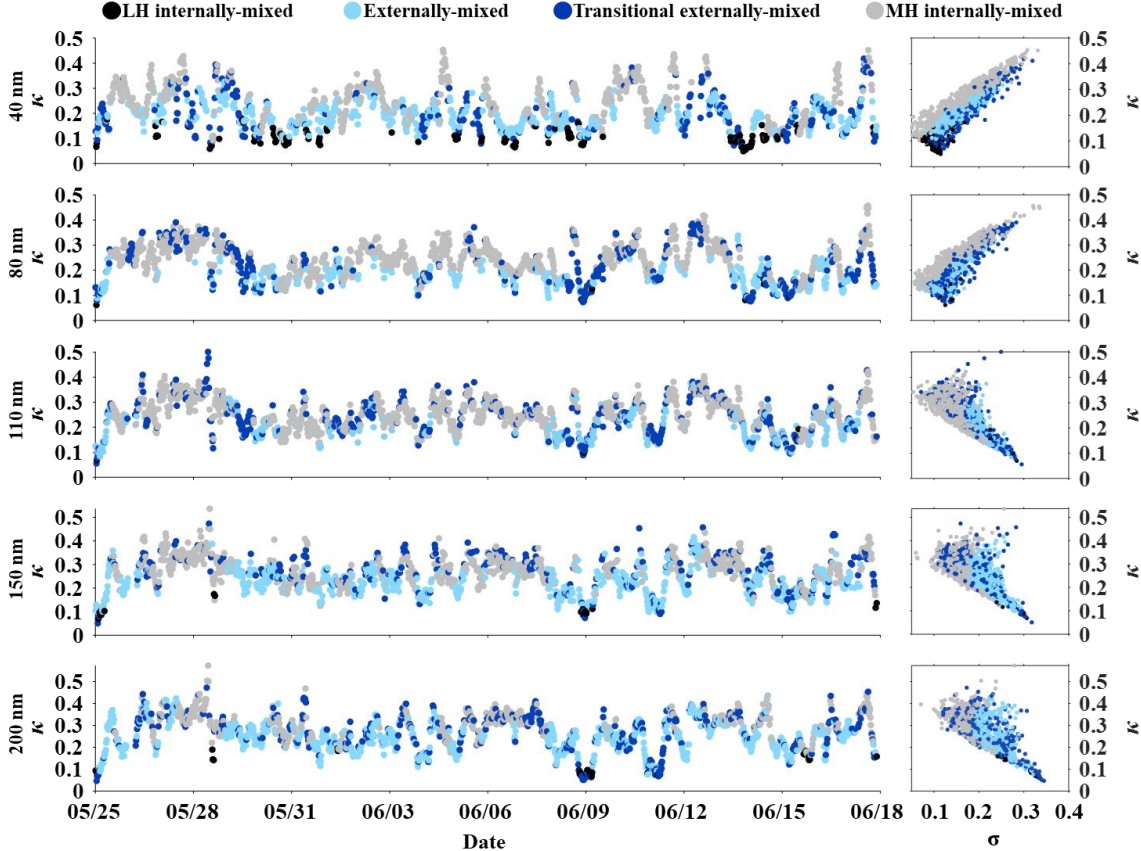


Figure 2. Time series of hygroscopic parameter ($\kappa$), and plots of $\kappa$ versus standard deviation
of $\kappa$-PDF (σ) for 40, 80, 110, 150 and 200 nm particles (from top to bottom). The colors
denote the different mixing types.
The size dependence of mixing type fractions and σ are shown in Fig. 3. The external
mixing types, including externally-mixed and transitional externally-mixed states, dominate
in 40, 150 and 200 nm particles, accounting for 53%, 57% and 70% respectively of all
mixing types. The elevated fraction of externally-mixed particles is intimately associated with
the local sources. For instance, a previous study revealed the particle size distribution of
aerosols from different sources in urban Beijing, suggesting that the particles with diameter
of 40 nm during rush hours and cooking times predominantly originated from primary
emissions, and particles at 150 and 200 nm displayed a strong correlation with regional
atmospheric transport sources (Ren et al., 2023). In this study, the particles exhibit weaker
hygroscopicity at lower wind speeds, indicating that hydrophobic particles are primarily from
local sources (Fig. S3). Furthermore, for 40 nm particles, the relatively higher fraction of MH
internally-mixed particles (37%) is likely due to the growth and aging of newly formed
particles (Liu et al., 2021a). This is similar to the result observed in Athens, Greece, in which
the particles with 30 nm were more internally-mixed (Spitieri et al., 2023). For particles with
sizes of 80 and 110 nm, the transitional externally-mixed and MH internally-mixed states
totally account for 50-80% of all mixing types, corresponding to a significant decrease from
about 10% to less than 1% of the fraction of the LH internally-mixed type. Note that the
proportion of transitional externally-mixed particles remain relative constant, with a mean
value of 21% across all sizes in this study, implying a continuous influence of atmospheric
aging process on the particles mixing state. In addition, the larger particles (i.e., 150 nm and
200 nm) are closely associated with the growth of smaller particles and undergo a prolonged
aging process. However, the fraction of MH internally-mixed particles at 150 nm and 200 nm
is lower than that of smaller particles (e.g., 80 nm and 110 nm). This is mainly due to that a
considerable amount of accumulated particles observed during the campaign were emitted
from primary sources, which could be externally-mixed with most of the aged more
hygroscopic 150 and 200 nm aerosol particles. As shown in Fig. 3, the higher proportion of
externally-mixed state was obtained for particles with diameters of 150 nm and 200 nm.
Furthermore, the more active Brownian coagulation of smaller particles enhances chemical
homogenization (Park et al., 2002), resulting in a much higher fraction of MH internally-
mixed particles at 80 and 110 nm. On average, the result reveals that the externally and
internally mixed particles accounted for 53%±11% and 47%±11% respectively. The mixing
state of particles derived in this study differs from that reported by Zhang et al. (2017), in
which they conducted the measurements at a suburban site of Xinzhou, where the aerosols
are much less affected by sources nearby, and the aerosols are mainly transported from
elsewhere and are thus more aged and well mixed. The result implies that the influence of
mixing state on hygroscopicity should be explored at specific particle sizes due to the
heterogeneity of chemical compositions with particle size (Fan et al., 2020, Fig. S4).

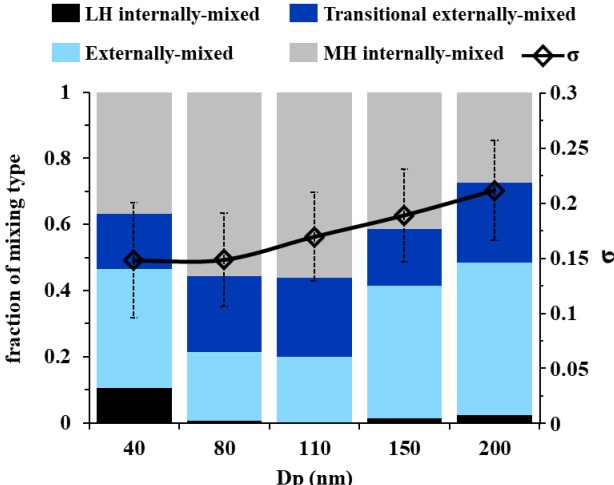

Figure 3. The size dependence of fraction of particles mixing types and σ.

Further, we compared the dependence of $\kappa$ and σ on the variations of mixing states for

five particle sizes (Fig. 4). At 40 and 80 nm sizes, the transition from externally-mixed to MH
internally-mixed particles results in a marked increase in $\kappa$, accompanied by a slight rise in σ.
Conversely, for particles with sizes of 110, 150 and 200 nm, the σ reduced but the $\kappa$ increased
when the particles change from externally to MH internally-mixed state. In other words, the σ
of the particles larger than 80 nm shifts to lower values as the particles become more

internally mixed. This contrasts with that previously observed in a vegetated site by Spitieri et al. (2023), which used only σ of Gf-PDF as a single indicator to characterize the particles mixing state. The distinct relationship between mixing type and σ for small particles (i.e., 40 and 80 nm) in urban sites indicates that the standard deviation of $\kappa$-PDF can effectively characterize mixing degree and chemical composition heterogeneity of larger particles in polluted area, where smaller aerosol particles are usually severely affected by local anthropogenic emissions.

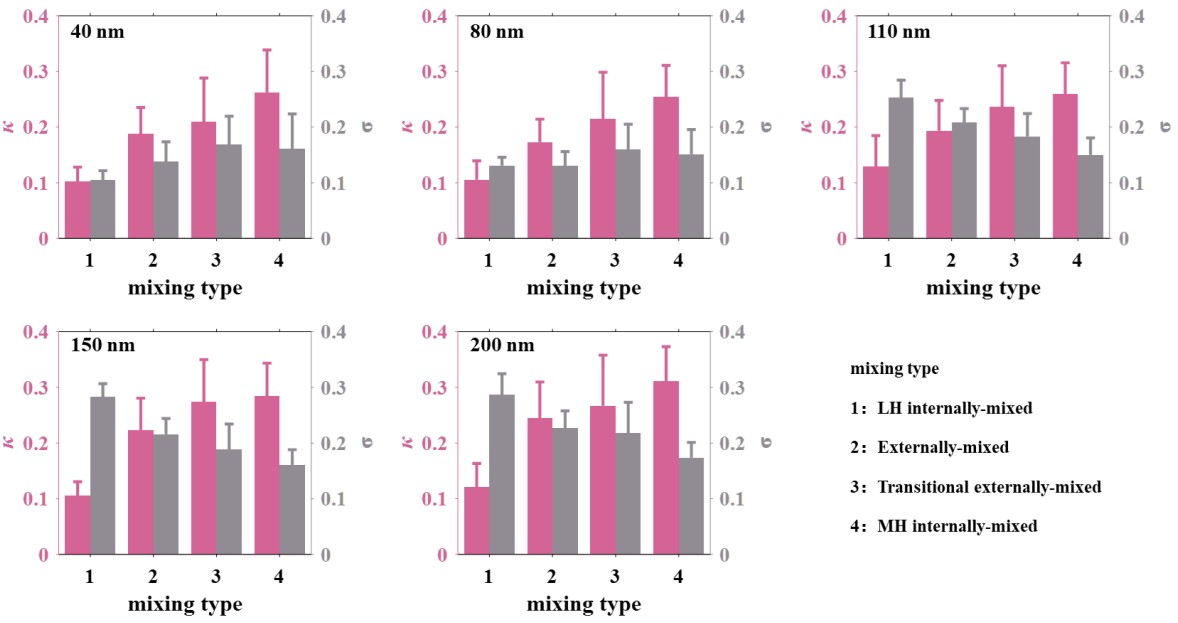

Figure 4. The mixing type dependence of hygroscopicity and σ for five particle sizes.

## 3.2 Evolution of mixing state of the particles

The average diurnal variations of the mixing state and $\kappa$ of different particle sizes are shown in Fig. 5. For 40 nm particles, the fraction of LH internally-mixed particles presents three peaks at morning (9:00–12:00 local time; LT), evening rush hours (18:00–20:00 LT) and nighttime (0:00–03:00 LT). This is accompanied with the impacts from those primary cooking and traffic emissions (Xu et al., 2021; Liu et al., 2021b). Unlike the 40 nm particles, there is no apparent increase of LH internally-mixed state for the particles with sizes of 80, 110, 150, and 200 nm in the rush hours or cooking times. The results indicate that the

particles emitted from local primary sources are small mainly with sizes around 40 nm during
the campaign. Correspondingly, the proportion of the particles with MH internally-mixed
state was smallest in the morning and nighttime, but exhibiting a rapid increase from about
9:00 until the evening rush hours (about 16:00–18:00). This is accompanied by the increase
of hydrophobic modes in $\kappa$-PDFs of small particle sizes during rush and cooking hours
measured in Beijing (Fig. S5). Moreover, the increase of externally-mixed particles of 40 nm
at nighttime may be highly associated with the emissions of primary species from heavy
trucks in urban Beijing at nighttime (Hua et al., 2018), which can also be evidenced by the
increased hydrophobic mode of the 150 nm and 200 nm particles observed from 00:00 to
around 3:00 in Fig. S5. In addition, the absence of photochemical aging and the lower
boundary layer height during nighttime could also result in a higher proportion of externally-
mixed particles. In the early afternoon, the externally-mixed and transitional externally-mixed
particles, which represent the intermediate state of the aging process in which particles
transition from LH to MH mode, showed a decrease to around 10%, while the proportion of
particles with MH internally-mixed state increased up to 80%. As a result, an obvious
enhancement in particles hygroscopicity was observed during the daytime, indicating the
impact of particles mixing and aging on their hygroscopicity (Hersey et al., 2013; Müller et
al., 2017). Overall, the diurnal variation implies an apparent aging process that leads the
particles to change from externally-mixed in the early morning to MH internally-mixed in the
afternoon.

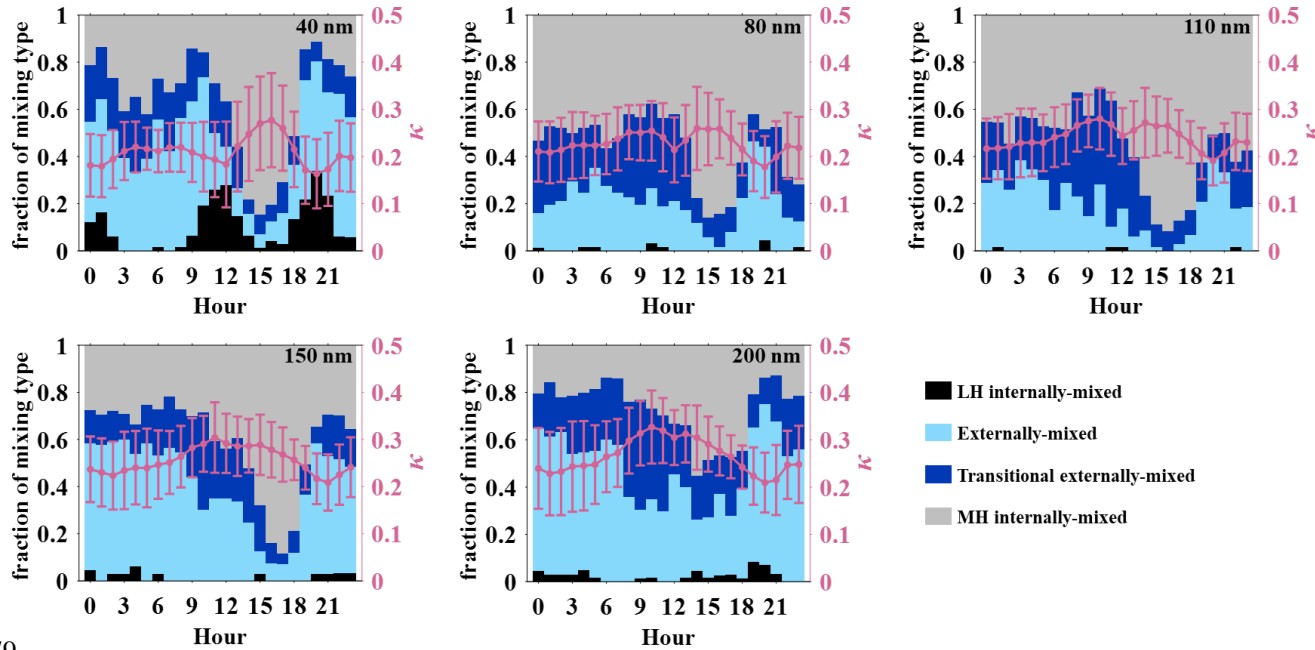


Figure 5. Diurnal variations of the mixing state fraction and $\kappa$ for five particle sizes.
**3.3 Particles mixing (aging) timescale: on clear and cloudy days**
The aerosol particles mixing and aging process is usually accompanied by the growth of
particle sizes. In order to derive the mixing and aging timescale of particles, we further
analyzed the averaged diurnal variations of particles fraction with MH internally-mixed state,
particle number size distributions (PNSD) and the variations of mean $\kappa$-PDF during the
growth periods on both clear and cloudy days (Fig. 6). On clear days, the fraction of MH
internally-mixed particles increases significantly from ~10-40% before 9:00 to ~70-100%
during 12:00–15:00. Such enhancement is nearly as large as that on cloudy days. In detail, the
proportion of MH internally-mixed particles usually increased by less than 50% between the
period before 9:00 and during 12:00–17:00, and its maximum fraction was around 50%-90%.
In addition, the fraction of MH internally-mixed particles for 40 nm is noticeably higher on
clear days (80%), which is approximately 40% greater than that on cloudy days, indicating
that photochemical processes can make particles more internally-mixed and aged (Liu et al.,
2021a). Furthermore, the diurnal variations of PNSD show that the growth of particle size is
highly consistent with the increase of the fraction of MH internally-mixed particles.
Specifically, as the particle size grows from less than 40 nm before 9:00 to ~100 nm around
17:00, the fraction of MH internally-mixed particles increases from less than 40% to >90%.
And, the peak value for $\kappa$ shifts from $\kappa < 0.1$ to $\kappa > 0.2$, and the count for the peak value of
$\kappa$-PDF increases accordingly (Fig. 6). Overall, on clear days, the particles undergo a gradual
shift from externally-mixed to MH internally-mixed states during 8:00–16:00 accompanied
by a growth in particle size from 20 nm to about 100 nm, which is generally 1–2 hours
shorter than that observed on cloudy days.

The obvious banana-shape growth of newly formed particles was captured in the diurnal

variations of PNSD, which characterizes that the aging and growth of non-hygroscopic
particles (e.g., BC) of primary emission and newly generated particles. Overall, the mixing
and aging process is confirmed by the transition of particle from externally-mixed to MH
internally-mixed, as well as the growth of particle size, which typically spans a duration of
approximately 5 to 10 hours, as is similar to the aging timescale of aerosol particles in the
polluted Indo-Gangetic Plain (< 10 hours) (Ghosh et al., 2021). In addition, the location of air
parcels 5–10 hours prior to measurement was close to the sampling site in Beijing, showing
that the air parcels from the northwest (cluster 5) with long-distance transport account for the
least fraction (only 8%) of the air trajectories from all originating directions during the
campaign (Fig. S1). This implies that most MH internally-mixed particles were formed
through the aging of locally emitted externally-mixed particles. Actually, the aerosol particles
aging would be largely affected by local atmospheric conditions, and thus would vary both
spatially and temporally (Pöschl et al., 2001; Huang et al., 2013). For example, using an
environmental chamber approach, Peng et al. (2016) revealed that the timescale of BC
aerosols aging is 2.3 and 4.6 hours, 9 and 18 hours over two cities—Beijing and Houston
respectively. Note that the faster aging time in Beijing derived in the chamber experiment is
probably due to the different levels in the concentration of gaseous precursors. In addition,
there is only photochemical aging occurred in the chamber experiment, however, in the
atmosphere, the particles aging is also through coagulation process which usually occurs
slower than the photochemical reaction and condensation processes (Chen et al., 2017). This
can also explain our observed faster mixing (aging) time on clear days when the
photochemical process is more significant considering that the difference of the pollutant
concentrations in daytime between clear and cloudy days is not obvious (i.e., $O_X$ and $PM_1$;
Fig. S6).

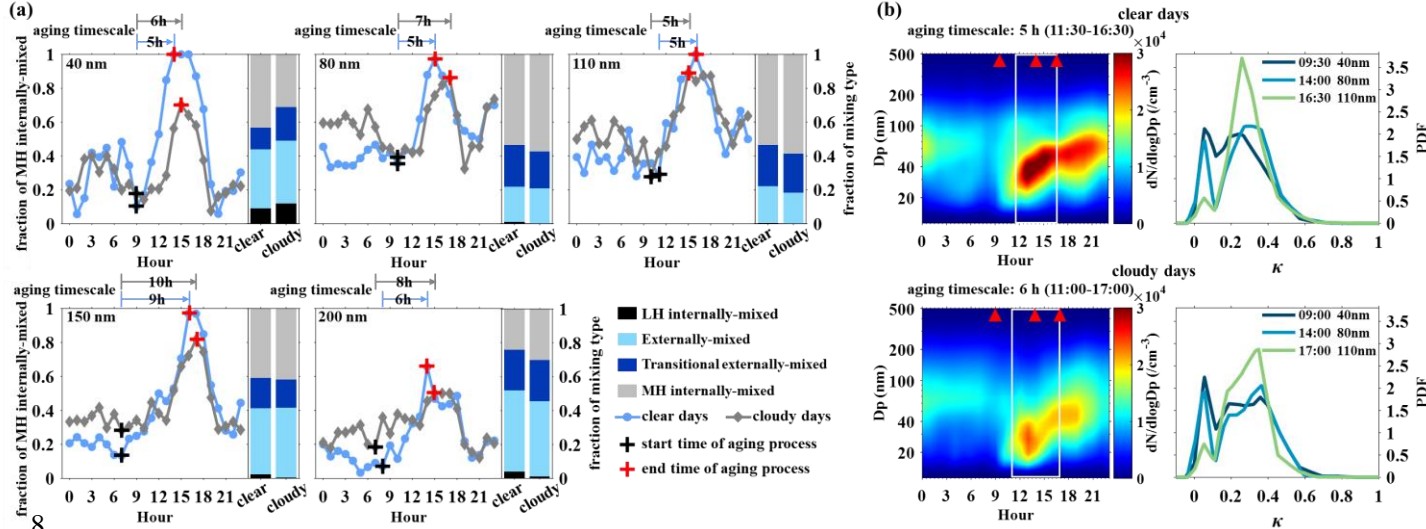

Figure 6. (a) Diurnal variations of the proportion of MH internally-mixed particles and
statistical results of the fraction of mixing state for five particle sizes on clear and cloudy
days. The start time of particle mixing/aging was selected when the proportion of MH
internally-mixed particles is minimum between sunrise and noon, and the end time was
defined as when the fraction of MH internally-mixed particles reaches its maximum. (b) The
averaged diurnal variations in PNSD and the variations of mean $\kappa$-PDF during the growth
periods on clear and cloudy days (right panels). The red triangles indicate 2 hours before the
start of the aging, during the growth and end of the aging process respectively.
**3.4 Implications to parameterization of the current models**
The mixing/aging timescale of aerosol particles significantly impacts their
physiochemical properties, which in turn affects their atmospheric lifetime, transport
characteristics (Zhang et al., 2023), hygroscopicity and the direct radiative forcing of aerosol
particles (Moffet and Prather, 2009; Wang et al., 2018; Stevens and Dastoor, 2019). We
further compared our results with the results derived from five field sites and the values
adopted in current models (Table S1). In most models, the aging processes of carbonaceous
aerosols were treated with a simple parameterization, which generally obtained aging time
based on the transformation of aerosols from hydrophobic to hydrophilic (Chen et al., 2017;
Ghosh et al., 2021). Unlike the models, the aging timescale of BC aerosols in field
measurements and laboratory studies was commonly derived by tracking changes in
hygroscopicity, morphology, coating thickness or optical properties (Moffet and Prather, 2009;
Akagi et al., 2012; Krasowsky et al., 2016; Peng et al., 2016). These methods were used to
quantify the aging timescale of particles indirectly. However, in our study, the mixing (aging)
time was inferred based on the changes in hygroscopic modes and mixing states of non-BC
particles in the ambient atmosphere. Note that although there may be some differences in the
results based on different methods, the aging timescales obtained all represent the mixing and
aging rate of aerosol particles in the atmosphere, which affects aerosols atmospheric lifetime,
thus the environment and climate effects. As shown in Fig. 7, the derived aging timescale of
aerosols in this study is comparable to the observational results reported in other urban areas,
such as Mexico City (3 hours; Moffet and Prather, 2009) and Los Angeles (3 hours;
Krasowsky et al., 2016). The result is also close to the aging timescale of particles in source
area of biomass burning in California (4 hours; Akagi et al., 2012). However, the aging time
of particles displays large spatial variations at different sites. For example, the aging
timescale of particles observed in Beijing (4.6 hours) was four times faster than that in
Houston (18 hours) where the precursors concentrations are extremely low (Peng et al., 2016).
Overall, the mixing/aging timescale of particles in Beijing achieved in this study is 5–10
hours, which falls within the range of 2–10 hours in polluted areas obtained by other studies.
Therefore, our results are likely representative of the aging time scale of aerosols in polluted
urban areas. Furthermore, the aging time of particles obtained in ambient atmosphere is much
shorter than the default values adopted in most models, which is commonly with a duration of
1.15–2.5 days. In addition, the values among different models range greatly from 1 to 20 days
(Fig. 7). For example, a timescale of 20 days was used to represent a slow aging process (i.e.,
coagulation) by Liu et al. (2011), as may be not properly applied in regions with high particle
number concentration where the particles coagulation is also efficient (Chen et al., 2017).
Although using a dynamic parameterization scheme for particles aging in models could
achieve the application of different aging times to different regions, the simulated values
show large variations among different models. For example, in RegCM4 model, the
conversion time from fresh to aged BC ranges from about 5 hours to 7 days (Ghosh et al.,
2021). The range in the KAMM/DRAIS, however, is only 2 hours to about 1.6 days (Riemer
et al., 2004). Moreover, even in central-eastern China, the aging timescale has been reported
to range from 12 hours to 7 days based on a regional chemical transport model (Chen et al.,
2017), which is much longer than that derived in urban Beijing by this study. Additionally,
the aging times of carbonaceous aerosols simulated by the GEOS-Chem model exhibit large
spatial and temporal variations, with the global average calculated to be 3.1 days (Huang et
al., 2013). This value is much longer than the default values used in other global models (1-2
days) (Pierce et al., 2007). Note that although the mixing (aging) timescale of particles was
determined by examining diurnal variations in the mixing state of aerosol particles in this
study, compared with laboratory studies, various factors in the real ambient atmosphere
change over time, including meteorological conditions, emission sources, and atmospheric
processes etc. Therefore, the result derived in this study warrants further verifications. The
long-term field measurements at more sites should be conducted in the future.

The large uncertainties in the aging timescale can significantly affect the accuracy of

simulation and assessment for atmospheric lifetime, loading and radiation forcing of aerosols.
For example, to better simulate the intercontinental transport of aerosols, Huang et al. (2013)
implemented a variable aging scheme in the GEOS-Chem model, which showed that the total
atmospheric burdens and global average lifetimes of BC (OC, organic carbon) were increase
by 8% (2%) compared to the default value (1.15 days). Similarly, due to the implementation
of the dynamic aging scheme in models, the column burden and surface mass concentration
of carbonaceous aerosols increased during the dry season in the polluted Indo-Gangetic Plain,
and the atmospheric heating increased by at least 1.2 W m$^{-2}$ (Ghosh et al., 2021). Therefore,
given the large spatiotemporal variations in aging timescale of particles, the study emphasizes
the urgency of conducting investigations at more field sites. In addition, the other factors such
as the meteorology and particle sizes that affect the aerosols aging should be accounted for so
as to improve the dynamic aging schemes in climate models.

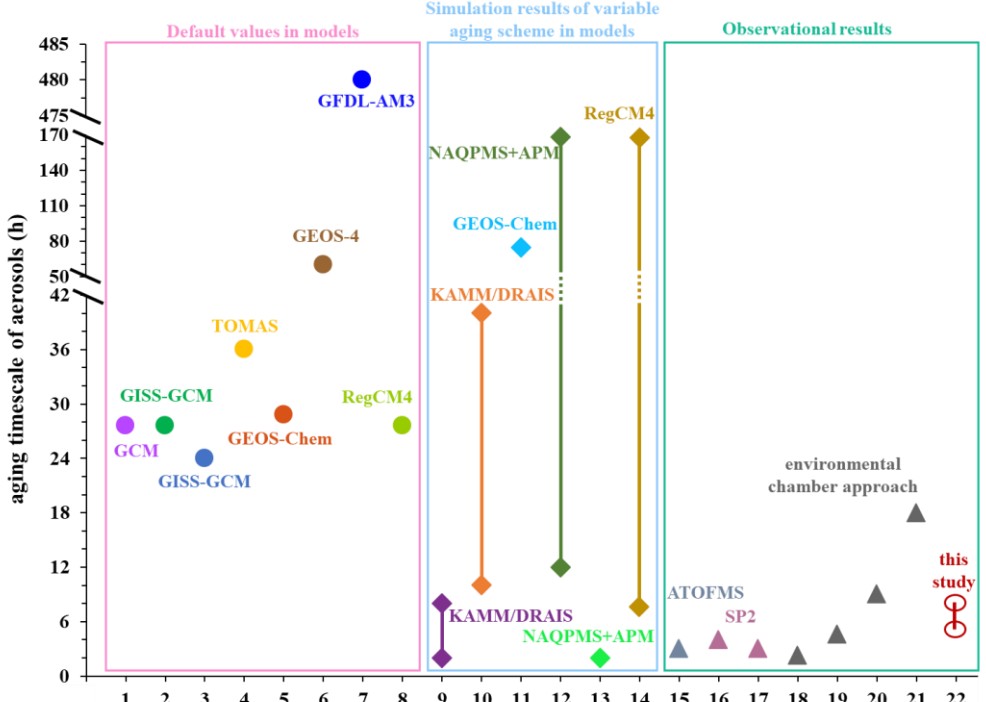

Figure 7. The mixing/aging timescale of particles reported in literatures (1. Cooke et al., 2002;

2. Chung and Seinfeld, 2002; 3. Koch and Hansen, 2005; 4. Pierce et al., 2007; 5. Yu and Luo,

2009; 6. Colarco et al., 2010; 7. Liu et al., 2011; 8, 14. Ghosh et al., 2021; 9, 10. Riemer et al.,

2004; 11. Huang et al., 2013; 12, 13. Chen et al., 2017; 15. Moffet and Prather, 2009; 16.

Akagi et al., 2012; 17. Krasowsky et al., 2016; 18-21. Peng et al., 2016). The solid circle,

diamond and the triangle denote the default aging time of particles used in models and

stimulation results of variable aging scheme in models, as well as the observational results,

respectively.

**4 Conclusions**

The real-time mixing state of ambient aerosol particles with dry particle sizes of 40, 80,

110, 150 and 200 nm was investigated in urban Beijing, according to PDF of hygroscopic

growth factor measured using the H-TDMA system. Four mixing states of ambient size-

resolved particles were captured in this study. In general, particles with LH internally-mixed,

externally-mixed, transitional externally-mixed and MH internally-mixed state account for 0-10%, 20-46%, 17-24% and 27-56% respectively, which depends on particles size greatly. The fraction of MH internally-mixed particles peaks at 80 and 110 nm due to more active Brownian coagulation enhances the chemical homogenization of particles. The diurnal variation of mixing state of particles in all sizes considered presents a visible aging process, showing that the fraction of particles with MH internally-mixed state increases significantly from ~10-40% before 9:00 to about 100% during 12:00–15:00 on clear days, accompanied by a growth in particle size from 20 nm to 100 nm. Typically, the mixing/aging process of particles takes approximately 5–10 hours, which reveals the mixing (aging) timescale of aerosols during the campaign. Additionally, our results suggest that fine aerosol particles undergo significant aging through photochemical processes in the polluted atmosphere of urban Beijing. Furthermore, there is a large difference in the mixing/aging timescale of particles between values in models and the timescale achieved by observations, emphasizing the vital role of exploring the aging timescale through more field measurements to improve the accuracy of aging schemes in climate models. The results revealed in our study highlight the considerable impact of atmospheric aging on mixing state of fine aerosol particles in polluted megacities.

**Data availability**

All data used in the study are available from the corresponding author upon request (zhangfang2021@hit.edu.cn).

**Author contributions**
FZ and JL conceived the conceptual development of the paper. FZ, JR, LC and JL
directed and performed of the experiments with AZ, ZW, SZ, HX and XY. JL conducted the
data analysis and wrote the draft. All authors edited and commented on the various sections
of the paper.
**Competing interests**
The authors declare that they have no conflict of interest.
**Acknowledgments**
This work was funded by the National Natural Science Foundation of China (NSFC)
research project (Grant No. 42475112, 42405191), Shenzhen Science and Technology Plan
Project (Grant No. GXWD20220811174022002; KCXST20221021111404011), Guangdong
Natural Science Foundation (Grant No. 2024A1515011005), Hebei Natural Science
Foundation (Grant No. D2024205006). We thank all participants in the field campaigns for
their tireless work and cooperation.

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
