# Peer review of "The evolution of aerosols mixing state derived from a field campaign in"

_EGUsphere, 2024_

## Author Response (AR2)

A point-by-point response to reviewers

Dear Editor,

We would like to submit our revised paper entitled "The evolution of aerosols mixing state derived from a field campaign in Beijing: implications to the particles aging time scale in urban atmosphere". We are grateful to the two reviewers for their insightful and constructive comments and have revised our paper accordingly to account for the reviewers' recommendations. Below please find our detailed point-by-point responses (in blue) to the reviewers' comments (in black) to the manuscript. We believe that we have addressed all concerns from the two reviewers.

Thank you for your attention to this matter.

Sincerely,

Fang Zhang

On behalf of all authors

Anonymous Referee #1

This study uses H-TDMA to identify in real-time the four mixing states of aerosols across different particle sizes, proposing changes in the transition from external to internal mixing under various conditions. Additionally, the study explains the possible causes and diurnal variations of the aging process associated with the transition of aerosol mixing states. The study finds significant differences between observations and models regarding aerosol aging timescales, suggesting that the models need improvements. This research offers innovative insights, providing valuable observational evidence for understanding the evolution of aerosols in the atmosphere. The study also reviews a large amount of observational and simulation results, which are highly useful for further research in this field.

When considering the mixing states of different particle sizes, how can we distinguish the impact of aging from smaller aerosols on the internal mixing fraction of larger particles, as well as the influence of primary emissions? Clarifying the relationship between these two factors will help better explain the causes of mixing state changes across different particle sizes.

Re: We really appreciate the invaluable comments and suggestions for this paper. The reviewer raised an important question, which is to distinguish the impact of aging of smaller particles and primary emissions on internal mixing fraction of larger particles. Given that the aerosol particles sampled at the campaign site in this study are more frequently affected by local traffic and cooking sources (Sun et al., 2015) and are rarely affected by other hygroscopic aerosols (e.g., marine aerosols). Therefore, it is considered that, the proportion of MH and LH internally-mixed particles at 100-200 nm can represent the contribution of the aging from smaller aerosols and the primary emissions to the internally-mixed larger particles, respectively. The statements have been added in the revised version, see **Lines 139-149** or as follows:

"Given that the aerosol particles sampled at the campaign site in this study are rarely affected by other hygroscopic aerosols (e.g., aerosols from marine sources), the MH internally-mixed particles (110, 150 and 200 nm) are thus mainly from the aging and growth of smaller particles. In contrast, the larger LH internally-mixed particles are predominantly from primary emissions. Therefore, the proportion of MH and LH internally-mixed particles at 100-200 nm can be used to distinguish the contribution to the internally-mixed fraction of larger particles from smaller aerosol aging and primary emissions respectively. Specifically, the result shows that a significant proportion

of the internally-mixed particles at larger size originate from the aging and growth of smaller particles, accounting for 99%, 96% and 92% respectively for 110, 150 and 200 nm particles, whereas the contribution from primary emissions to these larger internally-mixed particles is less than 10%."

The paper presents many interesting results; however, most of the results only present phenomena or conclusions without providing reasonable explanations or sufficient interpretation of the data, which may lead readers to question whether the results are justified.

Re: According to the reviewer's comments, we have provided additional explanations and interpretations to better justify our results. For example, in the revised manuscript, we have further addressed the particles aging and growth by including the diurnal variations of the particle number size distribution (PNSD) and the corresponding $\kappa$-PDF observed during the growth time periods (Figure 6). In addition, we have also included more discussions and statements in Section 3.4 regarding the difference in the aging timescale of aerosol particles as characterized by models and field measurements. See **Lines 282-289 and 328-340**, or as follows:

"Furthermore, the diurnal variations of PNSD show that the growth of particle size is highly consistent with the increase of the fraction of MH internally-mixed particles. Additionally, the particles $\kappa$ with large sizes at the corresponding times in the PNSD also shows an increase on both clear and cloudy days. Specifically, as the particle size grows from less than 40 nm before 9:00 to ~100 nm around 17:00, the fraction of MH internally-mixed particles increases from less than 40% to >90%. And, the peak value for $\kappa$ shifts from $\kappa < 0.1$ to $\kappa > 0.2$, and the count for the peak value of $\kappa$-PDF increases accordingly (Figure 6)."

"In most models, the aging processes of carbonaceous aerosols were treated with a simple parameterization, which generally obtained aging time based on the transformation of aerosols from hydrophobic to hydrophilic (Chen et al., 2017; Ghosh et al., 2021). Unlike the models, the aging timescale of BC aerosols in field measurements and laboratory studies was commonly derived by tracking changes in hygroscopicity, morphology, coating thickness or optical properties (Moffet and Prather, 2009; Akagi et al., 2012; Krasowsky et al., 2016; Peng et al., 2016). These methods were used to quantify the aging timescale of particles indirectly. While, in our study, the mixing (aging) time was inferred based on the changes in hygroscopic modes and mixing states of non-BC particles in the ambient atmosphere. Note that although there may be some differences in the results based on different methods, the aging timescales obtained all represent the mixing and aging rate of aerosol

particles in the atmosphere, which affects aerosols atmospheric lifetime, thus the environment and climate effects."

L137: You explain that σ cannot represent the mixing state of Beijing aerosols due to the influence of significant anthropogenic emissions. It would be helpful to provide a more detailed explanation of the relationship between anthropogenic emissions and mixing state, and the specific effects on σ and κ. You claim that σ alone is insufficient to characterize the mixing state of Beijing aerosols, but the relationship between the particle size distribution of the σ value and the vegetation-covered areas is not fully explained. Shouldn't you check whether the differences are due to methodological variations or regional characteristics?

Re: We appreciate your comments. In the revised version, we have provided explanations regarding the physical mechanisms of the σ-$\kappa$-mixing state relationships across different particle sizes. See **Lines 170-186** and **232-236** or as follows:

"The different sources and chemical compositions of aerosol particles could explain the distinct relationship between σ, mixing states and hygroscopicity among particles of different sizes. Here, the standard deviation of $\kappa$-PDF, σ, reflects the degree of dispersion in the pattern of $\kappa$-PDF. Taking 40 nm particles as an example, the obtained larger σ value indicates more diversity in particles hygroscopicity and chemical compositions, and thus a higher degree of external mixing state. This is because that, besides the local emissions (Ren et al., 2023), the 40 nm particles during the campaign were also from the growth of newly formed particles which are much more hygroscopic (Liu et al., 2021a). Therefore, the composition of those small particles is more heterogeneous, resulting in both greater σ and $\kappa$. The cases where both the σ and $\kappa$ of small particles are lower suggest that the particles may consist of a mixture of different species originating from primary emissions (e.g., BC, hydrocarbon-like and/or cooking organic aerosols). While, for most of larger particles, their chemical composition tends to become homogeneous after aging and growth processes, exhibiting stronger hygroscopicity, a higher degree of internal mixing, and thus the smaller σ. However, a small number of larger particles with weak hygroscopic properties from primary emissions coexist with the aged ones in the atmosphere, which would result in larger σ and higher degree of external mixing."

"The distinct relationship between mixing type and σ for small particles (i.e., 40 and 80 nm) in urban sites indicates that the standard deviation of $\kappa$-PDF can effectively characterize mixing degree

and chemical composition heterogeneity of larger particles in polluted area, where smaller aerosol particles are usually severely affected by local anthropogenic emissions."

L148: The sources of 150 nm and 200 nm aerosols mentioned here—are they primary emissions from traffic and biomass burning, or secondary aerosols? If they are from primary emissions, they should be similar to the 40 nm aerosols. Moreover, 80 and 110 nm aerosols should continue aging and form 150 and 200 nm aerosols. The aging of this group of aerosols should be more advanced— why is the influence of this aged aerosol group on the internal mixing fraction not reflected?

Re: Our previous version may have confused the reviewer. "Transportation sources" refers to long-range transport sources. The statement has been revised (see **Lines 195-199**) as follows,

"For instance, a previous study revealed the particle size distribution of aerosols from different sources in urban Beijing, suggesting that the particles with diameter of 40 nm during rush hours and cooking times predominantly originated from primary emissions, and particles at 150 and 200 nm displayed a strong correlation with regional atmospheric transport sources (Ren et al., 2023)."

According to reviewer's comment, we have included the statements in revised version, see **Lines 210-213**, or as follows:

"In addition, the 80 and 110 nm aerosols may continue aging and form 150 and 200 nm aerosol particles. Therefore, the aging of this group of aerosols should be more advanced, showing that the fraction of internally-mixed particles at 80 and 110 nm was the highest among the five particle sizes." L176: While the conclusion that the standard deviation of κ-PDF alone is insufficient to characterize the mixing state in polluted regions may be correct, could you further explain why the relationship between κ and σ exhibits contrasting characteristics for different particle sizes?

Re: According to this comment, we have added explanations for the different σ-*κ*-mixing state relationships between different particle sizes in the revised manuscript, see **Lines 170-186**, or as follows:

"The different sources and chemical compositions of aerosol particles could explain the distinct relationship between σ, mixing states and hygroscopicity among particles of different sizes. Here, the standard deviation of *κ*-PDF, σ, reflects the degree of dispersion in the pattern of *κ*-PDF. Taking 40 nm particles as an example, the obtained larger σ value indicates more diversity in particles hygroscopicity and chemical compositions, and thus a higher degree of external mixing state. This

is because that, besides the local emissions (Ren et al., 2023), the 40 nm particles during the campaign were also from the growth of newly formed particles which are much more hygroscopic (Liu et al., 2021a). Therefore, the composition of those small particles is more heterogeneous, resulting in both greater σ and κ. The cases where both the σ and κ of small particles are lower suggest that the particles may consist of a mixture of different species originating from primary emissions (e.g., BC, hydrocarbon-like and/or cooking organic aerosols). While, for most of larger particles, their chemical composition tends to become homogeneous after aging and growth processes, exhibiting stronger hygroscopicity, a higher degree of internal mixing, and thus the smaller σ. However, a small number of larger particles with weak hygroscopic properties from primary emissions coexist with the aged ones in the atmosphere, which would result in larger σ and higher degree of external mixing."

L185: Earlier, you mentioned that larger particles may be related to traffic and biomass emissions, but here you claim that the peak for 40 nm particles is related to traffic emissions, which seems inconsistent with previous statements.

Re: In fact, we pointed out that larger particles (i.e., 150 and 200 nm) have a strong correlation with regional atmospheric transport sources. While, particles with diameter of 40 nm during evening hours or cooking times predominantly originated from primary emissions (e.g., primary cooking and traffic emissions) (Xu et al., 2021; Liu et al., 2021a). In addition, the particles in nucleation mode also exhibit a significant dependence on the growth of newly formed particles during daytime (Liu et al., 2021b). The statement has been revised (see **Lines 195-199, 201-203**) as follows,

"For instance, a previous study revealed the particle size distribution of aerosols from different sources in urban Beijing, suggesting that the particles with diameter of 40 nm during rush hours and cooking times predominantly originated from primary emissions, and particles at 150 and 200 nm displayed a strong correlation with regional atmospheric transport sources (Ren et al., 2023)."

"Furthermore, for 40 nm particles, the relatively higher fraction of MH internally-mixed particles (37%) is likely due to the growth and aging of newly formed particles (Liu et al., 2021a)."

L190: If the large amount of external mixing in the early morning is due to fresh emissions, what is the source of the external mixing aerosols at night?

Re: Based on the comments and the diurnal variations in $\kappa$-PDF measured during this campaign (Figure R1), we have included additional statements in the revised manuscript (see **Lines 250-258**), as follows:

"This is accompanied by the increase of hydrophobic modes in $\kappa$-PDFs of small particle sizes during rush and cooking hours measured in Beijing (Figure S5). Moreover, the increase of externally-mixed particles of 40 nm at nighttime may be highly associated with the emissions of primary species from heavy trucks in urban Beijing at nighttime (Hua et al., 2018), which can also be evidenced by the increased hydrophobic mode of the 150 nm and 200 nm particles observed from 00:00 to around 3:00 in Figure S5. In addition, the absence of photochemical aging and the lower boundary layer height during nighttime could also result in a higher proportion of externally-mixed particles."

[Figure]

Figure R1. Campaign-averaged diurnal variations in $\kappa$-PDFs for all measured particle sizes (40–200 nm) at Beijing site.

L214: Is the proportion of internal mixing state reduced with increasing humidity? Does this suggest that aerosols undergo less aging in high humidity conditions? This appears to contradict the conclusion in line 217.

Re: Due to the strong correlation between temperature, relative humidity, $PM_1$ and the time of day, it is difficult to determine whether there are any independent associations between these factors and the mixing state or hygroscopic properties. Therefore, we have removed this section based on the comments from the reviewer 2 in the revised version.

L245: There is insufficient reliable evidence to support that this is a new particle formation process. I recommend not using this as a basis to explain the size characteristics of the mixing state.

Re: The statement has been revised in the main text (see **Lines 279-282**) or as follows,

L256: Clearly, the aging times of aerosols in different cities vary greatly. Given such large differences, how representative is your study?

Re: According to the reviewer's comment and the result shown in Figure R2, we have included some statements to illustrate this issue, see **Lines 348-351**, or as follows:

"Overall, the mixing/aging timescale of particles in Beijing achieved in this study is 5–10 hours, which falls within the range of 2–10 hours in polluted areas obtained by other studies. Therefore, our results are likely representative of the aging time scale of aerosols in polluted urban areas."

[Figure]

Figure R2. The mixing/aging timescale of particles reported in literatures (1. Cooke et al., 2002; 2. Chung and Seinfeld, 2002; 3. Koch and Hansen, 2005; 4. Pierce et al., 2007; 5. Yu and Luo, 2009; 6. Colarco et al., 2010; 7. Liu et al., 2011; 8, 14. Ghosh et al., 2021; 9, 10. Riemer et al., 2004; 11. Huang et al., 2013; 12, 13. Chen et al., 2017; 15. Moffet and Prather, 2009; 16. Akagi et al., 2012; 17. Krasowsky et al., 2016; 18-21. Peng et al., 2016). The solid circle, diamond and the triangle denote the default aging time of particles used in models and stimulation results of variable aging scheme in models, as well as the observational results, respectively.

L280: You point out that the observed aging time is similar to the aging times used in current models. Does this contradict the general conclusion that the aging timescale in models is inaccurately

represented? Furthermore, do simulation results show variations in aging timescales under different environmental and pollution conditions? Can these results be directly compared to observations? You may also consider including a range for the simulated aging times, as this comparison would be more scientifically rigorous.

Re: (1) The statement has been revised (see **Lines 340-343**) as follows:

"As shown in Figure 7, the derived aging timescale of aerosols in this study is comparable to the observational results reported in other urban areas, such as Mexico City (3 hours; Moffet and Prather, 2009) and Los Angeles (3 hours; Krasowsky et al., 2016)."

(2) The default aging timescale of particles and the results simulated by models are both shown in Figure R2 and Table R1. The results show that simulation results vary greatly among different regions. According to the reviewer's comment, we have included more statements in the revised version (see **Lines 358-369**), as follows:

"Although using a dynamic parameterization scheme for particles aging in models could achieve the application of different aging times to different regions, the simulated values show large variations among different models. For example, in RegCM4 model, the conversion time from fresh to aged BC ranges from about 5 hours to 7 days (Ghosh et al., 2021). The range in the KAMM/DRAIS, however, is only 2 hours to about 1.6 days (Riemer et al., 2004). Moreover, even in central-eastern China, the aging timescale has been reported to range from 12 hours to 7 days based on a regional chemical transport model (Chen et al., 2017), which is much longer than that derived in urban Beijing by this study. Additionally, the aging times of carbonaceous aerosols simulated by the GEOS-Chem model exhibit large spatial and temporal variations, with the global average calculated to be 3.1 days (Huang et al., 2013). This value is much longer than the default values used in other global models (1-2 days) (Pierce et al., 2007)."

Table R1. The aging timescale of particles reported in literatures.

| Region | Method | Aerosol type | Aging timescale | Aging indicators | Rreference |
|--------|--------|--------------|-----------------|------------------|------------|
| global | GCM model | carbonaceous aerosols | 1.15 days | hygroscopicity | Cooke et al., 2002 |

| | | | | | |
|---|---|---|---|---|---|
| (model default) | GISS-GCM model | carbonaceous aerosols | 1.15 days | hygroscopicity | Chung and Seinfeld, 2002 |
| | GISS-GCM model | BC | 1 day | hygroscopicity | Koch and Hansen, 2005 |
| | TOMAS model | carbonaceous aerosols | 1.5 days | hygroscopicity | Pierce et al., 2007 |
| | GEOS-Chem model | carbonaceous aerosols | 1.2 days | hygroscopicity | Yu and Luo, 2009 |
| | GEOS-4 model | carbonaceous aerosols | 2.5 days | hygroscopicity | Colarco et al., 2010 |
| | GFDL-AM3 model | BC | 20 days | the rate of hygroscopic aging is depended on the sulfuric acid gas concentration | Liu et al., 2011 |
| | RegCM4 model | carbonaceous aerosols | 1.15 days | hygroscopicity | Ghosh et al., 2021 |
| south-western Germany | KAMM/DRAIS model | soot | daytime: 2-8 h  nighttime: 10-40 h | mixing state | Riemer et al., 2004 |
| global | GEOS-Chem model | carbonaceous aerosols | 3.1 days | hygroscopicity | Huang et al., 2013 |

| | | | | | |
|---|---|---|---|---|---|
| central-eastern China | NAQPMS+APM model | BC | 12 h-7 days | hygroscopicity | Chen et al., 2017 |
| Beijing | | | 2 h | | |
| south Asia | RegCM4 model | carbonaceous aerosols | 7.6-167.6 h | hygroscopicity | Ghosh et al., 2021 |
| Mexico City | Field measurement ATOFMS | soot | 3 h | optical properties | Moffet and Prather, 2009 |
| California | Field measurement SP2 | BC | ~4 h | fraction of thickly coated rBC | Akagi et al., 2012 |
| Los Angeles | Field measurement SP2 | BC | ~3 h | fraction of thickly coated rBC | Krasowsky et al., 2016 |
| Beijing | Environmental chamber approach | BC | 2.3 h & 4.6 h | morphology & absorption amplification | Peng et al., 2016 |
| Houston | | | 9 h & 18 h | | |

(3) The reviewer suggests to consider including a range for the simulated aging times, as this comparison would be more scientifically rigorous. However, some studies only provided average or default values as shown in the summarized Table R1 used by the models. Note that although there may be some differences in the results based on different methods, the aging timescales obtained all represent the mixing and aging rate of aerosol particles in the atmosphere, which affects aerosols atmospheric lifetime, thus the environment and climate effects. We just have included some statements in the revised version, see **Lines 328-340,** or as follows:

[revised manuscript text omitted]

Anonymous Referee #2

In this manuscript, the authors show an analysis of H-TDMA measurements of aerosol in Beijing, and they attempt to identify the aging timescale of the transition of freshly emitted low-hygroscopicity aerosol to high-hygroscopicity aerosol and some factors that affect this aging timescale. The authors correctly identify this transition as a pertinent area of study, and they show some good insights, including that the standard deviation of the hygroscopicity yields different information that counting the number of hygroscopicity modes. However, there are some issues in the authors' analysis of the effects of temperature and relative humidity, their calculation of the aging timescale, and their comparison with other studies that require improvement before I can recommend this manuscript for publication. I will go into more detail on these issues below.

Re: We are grateful to the reviewer for your valuable comments and suggestions, all of which have been considered carefully during the revision (a point-by-point response to reviewer as follows).

There is unfortunately an existing ambiguity in the literature regarding the terms "externally mixed", "internally mixed", and "aging" or "aged". This has led to confusion, with different authors using different definitions and making comparisons across studies of different physical processes without identifying these differences. I therefore request that the authors make clear in the abstract and the conclusions that the aging timescales refer to hygroscopic aging. I also suggest that the authors adopt the terminology suggested by Riemer et al. (2019) regarding internal mixing and external mixing: these are properties of aerosol populations, rather than of individual aerosol particles. I strongly believe that the paper will be clearer and easier to understand if the authors are precise in their use of terms, and that the current terms leave an ambiguity that can result in reader confusion.

Re: We have clarified and added some illustrations for the term "mixing state" with reference to the description of Riemer et al. (2019), see **Lines 80-91**, or as follows:

   "According to Winkler's definition, the "internal mixing" means when all particles within the bulk aerosol populations are with the same compositions. While, the "external mixing" means when all particles consist of pure species and have distinct compositions (Winkler et al., 1973; Riemer et al., 2019). Note that in Riemer et al. (2019), the definition refers to the bulk not the sized-resolved mixing state of aerosol particles. In this study, the size resolved mixing state of aerosol particles has been identified according to the patterns of the probability density function of hygroscopic growth factor/hygroscopic parameter ($\kappa$) (Gf-PDF; $\kappa$-PDF) measured using a humidity tandem differential

mobility analyzer (H-TDMA) (Hong et al., 2018; Shi et al., 2022; Spitieri et al., 2023). For example, the internally-mixed aerosol populations with diameter of 40 nm are characterized by the single hygroscopic mode of Gf-PDF/$\kappa$-PDF, while the externally-mixed particles have bi- or trimodal distributions of Gf-PDF/$\kappa$-PDF."

In addition, we have clarified that the aging timescale retrieved in our study refers to mixing (aging) time. As shown in Figure R1 (or Figure 6), the aging time of the particles is clearly reflected by the aerosol populations of specific sizes changes from being dominated by externally-mixed to nearly completely more hygroscopic (MH) internally-mixed. This transition of mixing state of aerosol populations implies that the aging of particles undergo coagulation and/or other physicochemical processes that increase the particle sizes (Figure R1). Also, given that there is no simple linear relationship between the hygroscopicity and mixing state (Figure 2 and Figure 4 in manuscript), it is more precise to define the aging timescale of aerosol particles in our study as mixing (aging) time. Also, we have provided further explanations for figure 6 in the main text, see **Lines 282-289**, or as follows:

"Furthermore, the diurnal variations of PNSD show that the growth of particle size is highly consistent with the increase of the fraction of MH internally-mixed particles. Additionally, the particles $\kappa$ with large sizes at the corresponding times in the PNSD also shows an increase on both clear and cloudy days. Specifically, as the particle size grows from less than 40 nm before 9:00 to ~100 nm around 17:00, the fraction of MH internally-mixed particles increases from less than 40% to >90%. And, the peak value for $\kappa$ shifts from $\kappa < 0.1$ to $\kappa > 0.2$, and the count for the peak value of $\kappa$-PDF increases accordingly (Figure 6)."

[Figure]

Figure R1. (a) Diurnal variations of the proportion of MH internally-mixed particles and statistical results of the fraction of mixing state for five particle sizes on clear and cloudy days. The start time of particle mixing/aging was selected when the proportion of MH internally-mixed particles is minimum between sunrise and noon, and the end time was defined as when the fraction of MH internally-mixed particles reaches its maximum. (b) The averaged diurnal variations in particle number size distributions (PNSD) and the variations of mean $\kappa$-PDF during the growth periods on clear and cloudy days (right panels). The red triangles indicate 2 hours before the start of the aging, during the growth and end of the aging process respectively.

This is most notable in the naming of the "LH External Mixing" category: by the Riemer et al. (2019) nomenclature, the term "externally mixed" implies a population of aerosol particles with different chemical compositions. However, the hygroscopicity PDF shown in the leftmost panel of Fig. 1 is fairly uniformly non-hygroscopic. If the aerosol is assumed to consist of black carbon and low-hygroscopicity organic material, then the population may very well be internally mixed: each particle has the same composition of low-hygroscopicity material, consisting of black carbon partially or completely coated by low-hygroscopicity organic material. For the population to be externally-mixed, there need to be additional set(s) of aerosol particles with a different chemical composition. If the $\kappa$-PDF is to be the evidence of this, then there must be at least two peaks with different hygroscopicities. The "LH External Mixing" mode can certainly be described as low-hygroscopicity, but not as externally-mixed. Given the authors' measurement location, the smaller particles in this mode can likely be described as "LH freshly-emitted", although it is not difficult for me to think of processes (coagulation with other low-hygroscopicity particles, condensation/evaporation of low-hygroscopicity organic material) that could alter the particles without greatly increasing their hygroscopicity. To use this term, the authors may need to justify that the larger particles in this mode are not from long-range transport of biomass-burning aerosol or desert dust. Alternatively, it would be more precise but less evocative to term this "LH unimodal".

Re: Thanks for the comments. In the revised version, we have revised "LH external mixing" to "LH internally-mixed". Accordingly, the other 3 types have been revised as "Externally-mixed", "Transitional externally-mixed", and "MH internally-mixed", respectively (Figure R2). Also, the classification criteria for particle mixing states has been addressed in the main text, see **Lines 127-135**, or as follows:

"The $\kappa$-PDF of type 1 exhibits bimodal distributions with a much higher proportion (> 70%)

of hydrophobic mode or less hygroscopic mode (LH; the peak of $\kappa$-PDF occurs at $\kappa < 0.1$), was thus defined as LH internally-mixed state (type 1). Mixing type 2 refers to that the $\kappa$-PDF exhibits bimodal distributions in both LH and more hygroscopic (MH, $\kappa \geq 0.1$) modes, with the difference of less than 30% between the fraction of the two modes. The $\kappa$-PDF pattern with trimodal distributions was named as transitional externally-mixed state (type 3). The $\kappa$-PDF of type 4 was dominated by MH mode, normally with a fraction of lager than 70%, was defined as MH internally-mixed type."

[Figure]

Figure R2. The mean $\kappa$-PDF of five particle sizes and the schematic diagram of $\kappa$-PDF for four mixing types.

The ambiguity in the literature as to what constitutes "fresh" aerosol, "aged" aerosol, and what transition is being described by the "aging" timescale is especially pertinent for the comparison with the literature values of the aging timescale. In many of the models cited in Table S1 and Fig. 8, there is a convolution of the change in radiative properties (i.e. the absorption enhancement on thickly-coated black carbon vs. bare black carbon) and the change in the hygroscopic properties (low-hygroscopicity black carbon, primary organic aerosol, and dust becoming partially or completely coated in hydrophilic material) (Stevens and Dastoor, 2019). These two processes frequently are not represented independently, in part due to constraints on computational resources. However, the optical and hygroscopic properties do not necessarily co-vary; it is possible e.g. for black carbon to be thickly coated by low-hygroscopicity organic material, especially as this primary organic aerosol is frequently emitted by the same sources as black carbon. This ambiguity in "aging" exists in observational and experimental studies as well. I think that the authors are clear in referring exclusively to hygroscopic aging timescales for the modelling studies (despite the convolution I just described), and I appreciate the authors noting the measurement method for the experimental and observational studies mentioned in Table S1 and Fig. 8. However, I would also like them to make explicit what was the definition of "aged" used by each of the experimental or observational studies, especially regarding whether it was based on hygroscopicity, CCN activation, radiative properties,

or black carbon mass fraction. These different definitions will yield different aging timescales.

Re: The reviewer is right that there are differences in aging indicators of aerosols between models and observational studies. In detail, the aging processes of carbonaceous aerosols in most models were treated with a simple parameterization, which typically derived aging time based on the transformation of aerosols from hydrophobic to hydrophilic (Chen et al., 2017; Ghosh et al., 2021). While, the aging timescales of particles derived from field sites were usually quantified by changes in BC coating thickness or optical properties (Moffet and Prather, 2009; Akagi et al., 2012; Krasowsky et al., 2016; Peng et al., 2016). We have summarized the available results in literatures in Table R1 (Table S1) to make the comparison more scientific and rigorous. Additionally, we have added discussions on the differences in the aging timescales of aerosol particles characterized by models and field measurements, see **Lines 328-340**, or as follows:

"In most models, the aging processes of carbonaceous aerosols were treated with a simple parameterization, which generally obtained aging time based on the transformation of aerosols from hydrophobic to hydrophilic (Chen et al., 2017; Ghosh et al., 2021). Unlike the models, the aging timescale of BC aerosols in field measurements and laboratory studies was commonly derived by tracking changes in hygroscopicity, morphology, coating thickness or optical properties (Moffet and Prather, 2009; Akagi et al., 2012; Krasowsky et al., 2016; Peng et al., 2016). These methods were used to quantify the aging timescale of particles indirectly. While, in our study, the mixing (aging) time was inferred based on the changes in hygroscopic modes and mixing states of non-BC particles in the ambient atmosphere. Note that although there may be some differences in the results based on different methods, the aging timescales obtained all represent the mixing and aging rate of aerosol particles in the atmosphere, which affects aerosols atmospheric lifetime, thus the environment and climate effects."

Table R1. The aging timescale of particles reported in literatures.

| Region | Method | Aerosol type | Aging timescale | Aging indicators | Rreference |
|---|---|---|---|---|---|
| global | GCM model | carbonaceous aerosols | 1.15 days | hygroscopicity | Cooke et al., 2002 |

| | | | | | |
|---|---|---|---|---|---|
| (model default) | GISS-GCM model | carbonaceous aerosols | 1.15 days | hygroscopicity | Chung and Seinfeld, 2002 |
| | GISS-GCM model | BC | 1 day | hygroscopicity | Koch and Hansen, 2005 |
| | TOMAS model | carbonaceous aerosols | 1.5 days | hygroscopicity | Pierce et al., 2007 |
| | GEOS-Chem model | carbonaceous aerosols | 1.2 days | hygroscopicity | Yu and Luo, 2009 |
| | GEOS-4 model | carbonaceous aerosols | 2.5 days | hygroscopicity | Colarco et al., 2010 |
| | GFDL-AM3 model | BC | 20 days | the rate of hygroscopic aging is depended on the sulfuric acid gas concentration | Liu et al., 2011 |
| | RegCM4 model | carbonaceous aerosols | 1.15 days | hygroscopicity | Ghosh et al., 2021 |
| south-western Germany | KAMM/DRAIS model | soot | daytime: 2-8 h  nighttime: 10-40 h | mixing state | Riemer et al., 2004 |
| global | GEOS-Chem model | carbonaceous aerosols | 3.1 days | hygroscopicity | Huang et al., 2013 |

| central-eastern China | NAQPMS+APM model | BC | 12 h-7 days | hygroscopicity | Chen et al., 2017 |
|---|---|---|---|---|---|
| Beijing | | | 2 h | | |
| south Asia | RegCM4 model | carbonaceous aerosols | 7.6-167.6 h | hygroscopicity | Ghosh et al., 2021 |
| Mexico City | Field measurement ATOFMS | soot | 3 h | optical properties | Moffet and Prather, 2009 |
| California | Field measurement SP2 | BC | ~4 h | fraction of thickly coated rBC | Akagi et al., 2012 |
| Los Angeles | Field measurement SP2 | BC | ~3 h | fraction of thickly coated rBC | Krasowsky et al., 2016 |
| Beijing | Environmental chamber approach | BC | 2.3 h & 4.6 h | morphology & absorption amplification | Peng et al., 2016 |
| Houston | | | 9 h & 18 h | | |

There are additional problems in comparing the aging timescale derived in this study with those used in models and other studies: the authors are examining changes in hygroscopicity of a single size range, when the processes that would age the "LH External Mixing" mode from low-hygroscopicity to high-hygroscopicity, being coagulation and condensation, would also increase the size of the particles, potentially moving them from one size range to the next. Fresh nucleation and growth of ammonium-sulphate-nitrate-organic aerosol will create a mode of highly-hygroscopic aerosol; if the number of these freshly-nucleated particles greatly exceeds the number of "LH External Mixing" particles (possibly aided by the low-hygroscopicity particles growing out of this size range), then it will appear that the low-hygroscopicity particles have aged into a highly-hygroscopic particles, when few of these particles contain any low-hygroscopicity material. It appears to me in Fig. 5 and Fig. 7 that the peak in the fraction of internal mixing is later for the 110 and 150 nm particles than for the 40 and 80 nm particles, which may support growth of the

internally-mixed mode. There is also an issue of deriving lagrangian properties from single-location measurements: if the aerosol required e.g. six hours to become internally-mixed, then the "internally mixed" aerosol would have been emitted six hours prior to measurement. For an example wind speed of 10 km / hour, the "internally-mixed" aerosol would therefore have been emitted 60 km away, possibly with a different source profile than the "LH externally mixed" mode which is assumed to be freshly and locally-emitted. Additionally, the authors' method restricts the derived aging timescale to be less than 24 hours at maximum. The authors should at least discuss these difficulties in the interpretation of their aging timescale.

Re: We appreciate the valuable comments of the reviewer.

(1) Indeed, the aging process is usually accompanied by the growth of particle sizes. In order to verify the mixing and aging timescale of particles, we further analyzed the averaged diurnal variations of PNSD and the variations of mean $\kappa$-PDF during the growth periods on both clear and cloudy days. Figure 6 (Figure R1) in the main text has been updated in the revised version. More discussions on the particles mixing (aging) timescale have been included in section 3.3, see **Lines 269-299**, or as follows:

"**3.3 Particles mixing (aging) timescale: on clear and cloudy days**

The aerosol particles mixing and aging process is usually accompanied by the growth of particle sizes. In order to derive the mixing and aging timescale of particles, we further analyzed the averaged diurnal variations of particles fraction with MH internally-mixed state, particle number size distributions (PNSD) and the variations of mean $\kappa$-PDF during the growth periods on both clear and cloudy days (Figure 6). On clear days, the fraction of MH internally-mixed particles increases significantly from ~10-40% before 9:00 to nearly 100% during 12:00–15:00, especially for particles larger than 40 nm. Such enhancement is nearly as large as that on cloudy days. In detail, the proportion of MH internally-mixed particles usually increased by less than 50% between the period before 9:00 and during 12:00–17:00, and its maximum fraction was around 50%-90%. While, the fraction of MH internally-mixed particles for 40 nm is noticeably higher on clear days (80%), which is approximately 40% greater than that on cloudy days, indicating that photochemical processes can make particles more internally-mixed and aged (Liu et al., 2021a). Furthermore, the diurnal variations of PNSD show that the growth of particle size is highly consistent with the increase of the fraction of MH internally-mixed particles. Additionally, the particles $\kappa$ with large sizes at the

corresponding times in the PNSD also shows an increase on both clear and cloudy days. Specifically, as the particle size grows from less than 40 nm before 9:00 to ~100 nm around 17:00, the fraction of MH internally-mixed particles increases from less than 40% to >90%. And, the peak value for $\kappa$ shifts from $\kappa < 0.1$ to $\kappa > 0.2$, and the count for the peak value of $\kappa$-PDF increases accordingly (Figure 6). Overall, on clear days, the particles undergo a gradual shift from externally-mixed to MH internally-mixed states during 8:00–16:00 accompanied by a growth in particle size from 20 nm to about 100 nm, which is generally 1–2 hours shorter than that observed on cloudy days.

The obvious banana-shape growth of newly formed particles was captured in the diurnal variations of PNSD, which characterizes that the aging and growth of non-hygroscopic particles (e.g., BC) of primary emission and newly generated particles. Overall, the mixing and aging process is confirmed by the transition of particle from externally-mixed to MH internally-mixed, as well as the growth of particle size, which typically spans a duration of approximately 5 to 10 hours, as is similar to the aging timescale of aerosol particles in the polluted Indo-Gangetic Plain (< 10 hours) (Ghosh et al., 2021)."

(2) In addition, according to the reviewer's comments, we also have plotted a bivariate polar figure of the $\kappa$ of particles for five sizes during the campaign (Figure R3). It shows that, when the wind speed is lower than 4 m s$^{-1}$, the particles $\kappa$ for five sizes are smaller. That is, the particles exhibit weaker hygroscopicity at lower wind speeds, indicating that hydrophobic particles are primarily from local sources. We have made a statement in the revised version, see **Lines 199-201**, or as follows:

"In this study, the particles exhibit weaker hygroscopicity at lower wind speeds, indicating that hydrophobic particles are primarily from local sources (Figure S3)."

[Figure]

Figure R3. Bivariate polar plot of the $\kappa$ of particles at the five sizes during the campaign.

In fact, the aging timescale of particles is not limited to within 24 hours. One can examine 2-3 days data to look at the particles mixing, aging and growth based on the field campaign observation. However, during our campaign, a complete transformation process of particles from externally-mixed to MH internally-mixed was captured within a 24-hour period, in which the particles hygroscopicity was strengthened and their size was grown. These demonstrate that the mixing/aging timescale of aerosol particles in urban Beijing is within 24 hours. Based on the comments from the reviewer, we have added some statements in section 3.4, see **Lines 369-374**, or as follows:

"Note that although the mixing (aging) timescale of particles was determined by examining diurnal variations in the mixing state of aerosol particles in this study, compared with laboratory studies, various factors in the real ambient atmosphere change over time, including meteorological conditions, emission sources, and atmospheric processes etc. Therefore, the result derived in this study warrants further verifications. The long-term field measurements at more sites should be conducted in the future."

Given the strong association between temperature and time of day, it is difficult to tell if there are any independent associations between temperature and the mixing state or hygroscopic properties that are not best explained by diurnal cycles in emissions and the downward shortwave radiation

that drives photochemistry. Aerosol nucleation and condensation processes that drive hygroscopic aging can proceed more quickly under colder temperatures, not warmer temperatures, assuming that the actinic flux is constant. This criticism also applies to the analysis vs. relative humidity which also has a clear diurnal cycle. The authors therefore cannot draw any conclusions on the relationships between aging processes and temperature or relative humidity based on their current analysis. The authors should correct for this either by performing daily averages before analysing the data in this way or by removing the diurnal cycle before performing this analysis.

Re: As the reviewer pointed out, due to the strong correlation between temperature, relative humidity (RH), $PM_1$ and the time of day, it is difficult to determine whether there are any independent and meaningful associations between these factors and the mixing state or hygroscopic properties. Therefore, we have removed this section in the revision.

Specific comments:

Lines 97-99: The authors mention that the instrument was described previously. Was this particular measurement campaign described previously? Even if it was, the authors should still state where in Beijing the measurements were taken. On the roof of one of the authors' institutes? Co-located with an existing air pollution measurement site? Briefly, what is the source of the other measurements presented in this study (T, RH, PM1, chemical composition)? Were they measured by the authors or provided by another source? Additionally, can the presence of desert dust, especially in the larger size ranges, be excluded based on the chemical composition measurements? Or might long-range transport of dust be a significant contributor to the low-hygroscopicity particles?

Re: Here we have included some statements and descriptions of the site and the main instruments used in this campaign in the revised text, see **Lines 103-122**, or as follows:

"The campaign was conducted at the meteorological tower branch of the Institute of Atmospheric Physics (IAP), Chinese Academy of Sciences to measure the physical and chemical properties of particles and the meteorological conditions from 19 May to 18 June 2017 in Beijing. The instruments used in this study were deployed in a container at ground (the sampling inlet is located at ~8 m on the meteorological tower). The hygroscopicity of aerosols with different dry sizes (40, 80, 110, 150 and 200 nm) and particle number distribution in the size range of 10–550 nm was measured using a H-TDMA and a Scanning Mobility Particle Sizer (SMPS), respectively. The mass concentration of the non-refractory chemical compositions in particulate matter with diameter

< 1 μm (PM$_1$) was measured by an Aerodyne high-resolution time-of-flight aerosol mass spectrometer (HR-ToF-AMS) (Xu et al., 2019), and the BC was measured by a 7-wavelength aethalometer (AE33, Magee Scientific Corp). More details about the sites, instruments and calibrations can be found in previous literatures (Zhang et al., 2017; Wang et al., 2019; Fan et al., 2020; Chen et al., 2022). Since the campaign site was located at the urban area of Beijing, where the atmospheric fine aerosols are more frequently affected by local traffic and cooking sources (Sun et al., 2015). In addition, during the observation period of summertime, the site was predominately affected by the warm and humid southeastern air parcels (Figure S1); while the dust events normally originate from northwestern regions occurring in springtime (Li et al., 2020) and typically with particle sizes in coarse mode (Wang et al., 2023). Therefore, the contribution from dust aerosols could be negligible in this study."

line 25: "with the evolution of particle pollution…" It is not clear what the authors mean with this sentence. Are they merely saying that particles evolve from being externally-mixed to being internally-mixed with time?

Re: This statement has been removed in the main text.

Lines 76-79: "aerosols containing two or more components may also exhibit external mixing" Since external mixing is defined as different aerosol particles having different chemical compositions, the aerosol population must have two or more components for external mixing to occur.

Re: We have revised the statements (see **Lines 80-91**) as follows:

"According to Winkler's definition, the "internal mixing" means when all particles within the bulk aerosol populations are with the same compositions. While, the "external mixing" means when all particles consist of pure species and have distinct compositions (Winkler et al., 1973; Riemer et al., 2019). Note that in Riemer et al. (2019), the definition refers to the bulk not the sized-resolved mixing state of aerosol particles. In this study, the size resolved mixing state of aerosol particles has been identified according to the patterns of the probability density function of hygroscopic growth factor/hygroscopic parameter ($\kappa$) (Gf-PDF; $\kappa$-PDF) measured using a humidity tandem differential mobility analyzer (H-TDMA) (Hong et al., 2018; Shi et al., 2022; Spitieri et al., 2023). For example, the internally-mixed aerosol populations with diameter of 40 nm are characterized by the single hygroscopic mode of Gf-PDF/$\kappa$-PDF, while the externally-mixed particles have bi- or trimodal distributions of Gf-PDF/$\kappa$-PDF."

Lines 107-109: Should I understand the authors to mean that there must be two hygroscopicity modes, and one of the mode peaks must be at κ < 0.1 and the other mode peak at κ > 0.1? Does it matter which of the two peaks is greater? This was not clear to me.

Re: We have revised the statements (see **Lines 127-135**) as follows:

"The $\kappa$-PDF of type 1 exhibits bimodal distributions with a much higher proportion (> 70%) of hydrophobic mode or less hygroscopic mode (LH; the peak of $\kappa$-PDF occurs at $\kappa < 0.1$), was thus defined as LH internally-mixed state (type 1). Mixing type 2 refers to that the $\kappa$-PDF exhibits bimodal distributions in both LH and more hygroscopic (MH, $\kappa \geq 0.1$) modes, with the difference of less than 30% between the fraction of the two modes. The $\kappa$-PDF pattern with trimodal distributions was named as transitional externally-mixed state (type 3). The $\kappa$-PDF of type 4 was dominated by MH mode, normally with a fraction of lager than 70%, was defined as MH internally-mixed type."

Lines 109-110: Is it required that one of the modes in the trimodal category have κ < 0.1? Further, if there are 2 or 3 modes where all of the peaks have κ > 0.1, how would this be classified? Were there measurements that did not fit into any of the four categories?

Re: Thanks a lot for the comments. It actually was not required that the $\kappa$ value of one of the modes < 0.1 in the type 3 defined in our study. However, the measured pattern of $\kappa$-PDF for five particle sizes shows that the $\kappa$ value of one mode in bi- or trimodal modes is less than 0.1. It should be noted that the four categories defined in our study have covered all cases of the observational results. It is interesting that the reviewer figured out that if there are 2 or 3 modes of $\kappa$-PDF with all of the peaks have $\kappa > 0.1$ in the data, it could be classified as MH externally-mixed state according to this study. However, such cases may be observed when the site is influenced by more hygroscopic like sea salt sources.

line 122: I assume that the authors mean "in each of the five particle sizes". The sentence as written could also be interpreted as "between the five particle sizes".

Re: The sentence has been revised in manuscript (see **Lines 156-158**) or as follows:

"The $\kappa$ values vary from 0 to 0.5 in each of the five particle sizes, accompanied by significant changes in mixing states."

Lines 129-133: Do the authors have any thoughts as to a physical explanation for the different σ-κ relationships between different size ranges?

Re: According to this comment, we have included a physical explanation for the σ-$κ$-mixing state relationships between different particle sizes in the revised manuscript, see **Lines 170-186**, or as follows:

"The different sources and chemical compositions of aerosol particles could explain the distinct relationship between σ, mixing states and hygroscopicity among particles of different sizes. Here, the standard deviation of $κ$-PDF, σ, reflects the degree of dispersion in the pattern of $κ$-PDF. Taking 40 nm particles as an example, the obtained larger σ value indicates more diversity in particles hygroscopicity and chemical compositions, and thus a higher degree of external mixing state. This is because that, besides the local emissions (Ren et al., 2023), the 40 nm particles during the campaign were also from the growth of newly formed particles which are much more hygroscopic (Liu et al., 2021a). Therefore, the composition of those small particles is more heterogeneous, resulting in both greater σ and $κ$. The cases where both the σ and $κ$ of small particles are lower suggest that the particles may consist of a mixture of different species originating from primary emissions (e.g., BC, hydrocarbon-like and/or cooking organic aerosols). While, for most of larger particles, their chemical composition tends to become homogeneous after aging and growth processes, exhibiting stronger hygroscopicity, a higher degree of internal mixing, and thus the smaller σ. However, a small number of larger particles with weak hygroscopic properties from primary emissions coexist with the aged ones in the atmosphere, which would result in larger σ and higher degree of external mixing."

Lines 131-133: It's not clear to me what the authors are trying to communicate with the second half of this sentence. Are you simply trying to highlight that "internally-mixed" cluster for particles >100 nm in size has high hygroscopicity and low standard deviation in hygroscopicity?

Re: This sentence has been revised (see **Lines 164-170**) as follows:

"For particles with diameter of 40 and 80 nm, the σ increases as the $κ$ increases from 0.1 to 0.4 along with slight increase in the fraction of MH internally-mixed particles, showing a positive correlation. While, unlike the small particles, the σ for particles larger than 100 nm (i.e., 110, 150 and 200 nm) exhibits a negative correlation to $κ$ variations, showing that the particles were more MH internally-mixed and with stronger hygroscopicity when the smaller σ values were derived."

Lines 146-149: My reading of Ren et al. (2023) is that they associated transportation sources (traffic, in their words) with small particles (<100 nm), which is consistent with my previous understanding that vehicular emissions yield primary particles that are <100 nm. I don't recall Ren et al. (2023) discussing biomass burning at all. Do the authors have the correct reference here?

Re: This has been revised in the main text (see **Lines 195-199**), or as follows:

"For instance, a previous study revealed the particle size distribution of aerosols from different sources in urban Beijing, suggesting that the particles with diameter of 40 nm during rush hours and cooking times predominantly originated from primary emissions, and particles at 150 and 200 nm displayed a strong correlation with regional atmospheric transport sources (Ren et al., 2023)."

Lines 212-219: Looking at Fig. S4, 15 C temperatures seem to be rarely sampled, and possibly only during 3-6 am on clear days. Is it possible that all of the associations with 15 C temperatures are due to a sampling artifact, a small number of cases with coincidentally high numbers of hydrophilic aerosol and low temperatures? I note also that relative humidities > 60% seem only to occur during hours when temperatures are particularly low. Could a sampling bias also be affecting the relationships shown with relative humidity?

Re: We thank the reviewer for allowing us to consider this issue more carefully. Since the campaign was conducted in summertime, the frequency distribution histograms of ambient temperature (T), show that temperatures below 20°C were indeed rarely observed during the measurement period (Figure R4). In addition, the RH was mainly distributed between 20% and 80%, and particulate matter with diameter < 1 μm ($PM_1$) was mostly below 40 μg m$^{-3}$. As noted in the reviewer's previous comment, it is difficult to determine whether there are any independent and meaningful associations between these factors and the mixing state or hygroscopic properties, due to the strong correlation between temperature, relative humidity, $PM_1$ and the time of day. Therefore, we have removed this section in the revision.

[Figure]

[Figure]

[Figure]

Figure R4. The frequency distribution histogram of ambient T, RH and PM$_1$ during the campaign in Beijing.

line 239: While the enhancement is less for cloudy days, it is certainly notable. For particles larger than 40 nm, the enhancement on cloudy days is nearly as large as for clear days.

Re: Yes, the reviewer is right. We have revised the statements in the main text (see **Lines 274-282**), or as follows:

"On clear days, the fraction of MH internally-mixed particles increases significantly from ~10-40% before 9:00 to nearly 100% during 12:00–15:00, especially for particles larger than 40 nm. Such enhancement is nearly as large as that on cloudy days. In detail, the proportion of MH internally-mixed particles usually increased by less than 50% between the period before 9:00 and during 12:00–17:00, and its maximum fraction was around 50%-90%. While, the fraction of MH internally-mixed particles for 40 nm is noticeably higher on clear days (80%), which is approximately 40% greater than that on cloudy days, indicating that photochemical processes can make particles more internally-mixed and aged (Liu et al., 2021a)."

Lines 239-242: The first part of this sentence is confusingly written. I would suggest phrasing as "The proportion of particles with internal mixing state usually increased by less than 50 percentage points between the period before 9:00 and during 12:00–17:00".

Re: Thanks a lot for the comments. The sentence has been revised (see **Lines 277-279**) as follows:

"In detail, the proportion of MH internally-mixed particles usually increased by less than 50% between the period before 9:00 and during 12:00–17:00, and its maximum fraction was around 50%-90%."

Lines 331-332: This is an incomplete sentence.

Re: The sentence has been revised (see **Lines 409-410)** as follows:

"Typically, the mixing/aging process of particles takes approximately 5–10 hours, which reveals the mixing (aging) timescale of aerosols during the campaign."

Fig. 7: "The start time [...] of particle aging process was selected when the proportion of particles with internal mixing state is closest after sunrise". I am not sure what the authors mean by this. Do

the authors mean the hour when the proportion in the "internally mixed" state is most similar between the cloudy and clear cases? If so, then I don't understand why this was the criterion chose, as opposed to e.g. using the minimum between sunrise and noon, or using the maximum in this time range of the "LH externally mixed" state.

Re: Yes, the time when the proportion of MH internally-mixed particles is most similar between clear and cloudy days was selected as the start time of aging process.

According to the reviewer's suggestion, the start time of particle mixing/aging was selected when the proportion of MH internally-mixed particles is minimum between sunrise and noon, and the end time was defined as when the fraction of MH internally-mixed particles reaches its maximum. Then, we derive that the mixing/aging process typically spans a duration of approximately 5 to 10 hours.

[revised manuscript text omitted]

---

## Author Response (AR3)

**A point-by-point response to reviewers**

Dear Editor,

We are very pleased to submit a revised manuscript entitled with "The evolution of aerosols mixing state derived from a field campaign in Beijing: implications to the particles aging time scale in urban atmosphere" for possible publication in journal of Atmospheric Chemistry and Physics.

We'd like to thank you for your efforts and time on handling the paper. We also thank the two reviewers for the further comments, which we have addressed in the revision (a point-by-point response to the reviewer as follows).

Yours sincerely,

Fang Zhang

On behalf of all authors

**Comments from the reviewer 1:**

I appreciate the authors' responses to my comments and the substantial revisions made to the paper, which have significantly improved its quality. I believe it now largely meets the requirements for publication in ACP. The method proposed by the authors, using the proportions of MH and LH to distinguish between small particle growth and primary emissions, has a certain degree of rationality. However, it also involves notable uncertainties that may affect the conclusions of the paper. I suggest that the authors add some discussion on the uncertainties of this classification criterion and its impact on the conclusions.

Re: We appreciate your comments. In the revised version, we have included the statements regarding the uncertainties of the method used to distinguish the impacts of smaller particle aging and primary emissions on the internal mixing fraction of larger particles. See **Lines 149-155**, as follows:

"It is worth noting that these results may involve certain uncertainties, as the influence of other sources was not considered. For example, the more hygroscopic sulfate particles could also be directly emitted from primary sources in urban atmosphere (Dai et al., 2019); in addition, the aerosols are assumed rarely affected by the sea salts. Consequently, the contribution of smaller particle aging and growth to the internally-mixed fraction of larger particles may have been overestimated in this study. More research works are warranted to further evaluate and quantify such

uncertainty in future."

**Comments from the reviewer 2:**

I feel that the manuscript is greatly improved from its original version, and I appreciate the authors' efforts. However, I note one comment from the other reviewer and one of my own comments from the first round of review that I do not feel have yet been sufficiently addressed. Aside from these, I have only technical corrections to be addressed before publication.

Previously, reviewer # 1 commented that:

"80 and 110 nm aerosols should continue aging and form 150 and 200 nm aerosols. The aging of this group of aerosols should be more advanced—why is the influence of this aged aerosol group on the internal mixing fraction not reflected?"

I don't feel that the authors' addition to the manuscript (lines 210-213) addresses what the reviewer was trying to highlight. Perhaps they understood the reviewer differently than I did. If the same processes that move the particles from the other three mixing-state categories into the MH internally-mixed category also cause the growth of 80 and 110 nm aerosols into the 150 and 200 nm size ranges, one would expect that these particles would contribute to the MH internally-mixed category for the 150 and 200 nm size ranges, and not the 80 and 110 nm size ranges. So if these were the only important processes and all else was equal between the different size ranges, we would expect the MH internally-mixed fraction to be lower for the 80 and 110 nm particles than for the 150 and 200 nm particles. I therefore read the authors current lines 210-213 as being incongruous. Can the authors explain the apparent discrepancy?

Re: In our initial response, we may have misinterpreted the comments raised by Reviewer #1. As the reviewer pointed out, the 80 nm and 110 nm aerosols may continue to age and grow into 150 nm and 200 nm particles. However, the fraction of internally-mixed particles at 150 nm and 200 nm is lower than that of 80 nm and 110 nm particles. In the revised manuscript, we have clarified this point in **Lines 216-226**, as follows:

"In addition, the larger particles (i.e., 150 nm and 200 nm) are closely associated with the growth of smaller particles and undergo a prolonged aging process. However, the fraction of MH internally-mixed particles at 150 nm and 200 nm is lower than that of smaller particles (e.g., 80 nm and 110 nm). This is mainly due to that a considerable amount of accumulated particles observed during the campaign were emitted from primary sources, which could be externally-mixed with most of the aged more hygroscopic 150 and 200 nm aerosol particles. As shown in Fig. 3, the higher proportion of externally-mixed state was obtained for particles with diameters of 150 nm and 200

nm. Furthermore, the more active Brownian coagulation of smaller particles enhances chemical homogenization (Park et al., 2002), resulting in a much higher fraction of MH internally-mixed particles at 80 and 110 nm."

I previously noted the problem of determining Lagrangian properties from fixed-location measurements. I don't yet feel that the authors have adequately either addressed or acknowledged this issue. If the particles required 5-10 hours to age from the externally-mixed category into the MH internally-mixed category, can the authors assure the reader that the source emissions at the location of the air parcel 5-10 hours prior to measurement were similar to source emissions in Beijing? The back trajectories in Fig. S1 would help with this if the time length was indicated, as they would allow the reader an approximation of the location of the air parcels 5-10 hours before measurement. I thank the authors for sharing Fig. S3, but instead of reassuring the reader that MH internally-mixed particles are indeed externally-mixed particles that have aged, the figure could be interpreted as suggesting that for 40 nm and 80 nm particles, the externally-mixed and LH internally-mixed particles are coming from local sources and to the east, while MH internally-mixed particles are coming from a different source to the west.

Re: We appreciate the valuable comments of the reviewer.

(1) We fully acknowledge the challenges in inferring the historical behavior of air parcels from fixed-location measurements. According to the reviewer's comment, we have updated Fig. S1 (Fig. R1) in the revised version to clearly indicate the location of air parcels 5–10 hours before measurement, as follows:

[Figure]

Figure R1. The 48-hour mean air mass backward trajectories for five clusters arriving at the sampling site in Beijing. The black and red crosses indicate the locations of the air parcels 5 and 10 hours prior to the measurement site, respectively.

(2) As the reviewer pointed out, for 40 nm and 80 nm particles, the hydrophobic particles are primarily from local sources, whereas the hygroscopic particles originate from a different source to the west (Fig. S3 or R2). However, the particles exhibit weaker hygroscopicity at lower wind speeds, especially when the wind speed is lower than 4 m s$^{-1}$. Based on the statistical results during the observation period (Fig. R3), it showed that wind speeds below 4 m s$^{-1}$ accounted for 88% of all cases, indicating that the contribution of regional atmospheric transport sources to MH internally-mixed small particles was minimal. Most of MH internally-mixed particles were from the aging of externally-mixed particles.

To further clarify the origin of particles, we have updated Fig. S1 (Fig. R1) to include time markers, showing that the location of air parcels 5–10 hours prior to measurement was close to the sampling site in Beijing. The air parcels from the northwest (cluster 5) with long-distance transport account for the least fraction (only 8%) of the air trajectories from all originating directions during the campaign. This implies that most MH internally-mixed particles were formed through the aging of locally emitted externally-mixed particles.

We have included more statements to address this issue in the revised text, see **Lines 309-314**, or as follows:

"In addition, the location of air parcels 5–10 hours prior to measurement was close to the sampling site in Beijing, showing that the air parcels from the northwest (cluster 5) with long-distance transport account for the least fraction (only 8%) of the air trajectories from all originating directions during the campaign (Fig. S1). This implies that most MH internally-mixed particles were formed through the aging of locally emitted externally-mixed particles."

[Figure]

Figure R2. Bivariate polar plot of the $\kappa$ of particles at the five sizes during the campaign.

[Figure]

Figure R3. The frequency distribution histogram of wind speed during the campaign in Beijing.

Technical corrections:

p2, line 47: with -> in an

Re: Revised.

p4: line 72: "with the significance of statistics" do you mean statistically significant?

Re: This sentence has been revised as "However, based on TEM technique, a large number of aerosol samples are required so as to make the results statistically significant, which means a high cost both on labors and materials".

p4, line 81: are with -> have

Re: Revised.

p6, line 130: "refers to that the" please rephrase. Also, you should state the name of Mixing Type 2 here.

Re: The sentence has been revised as "Mixing type 2 (Externally-mixed) is characterized by bimodal distributions in the $\kappa$-PDF for both LH and more hygroscopic (MH, $\kappa \geq 0.1$) modes, with less than 30% difference in the fraction of the two modes".

p6, line 134: of lager -> larger. Also, was it only normally the case of >= 70% in the MH mode, or was it always the case, by definition, that >= 70% of the particles were in the MH mode?

Re: The sentence has been revised as "The $\kappa$-PDF of type 4, defined as the MH internally-mixed type, was dominated by the MH mode with a fraction larger than 70%".

p11, lines 225-226: Based on Fig. 4, this sentence appears to be untrue. κ increases, but for 40 and 80 nm particles, σ is similar (and certainly within the error bars shown) for mixing type 2 (externally-mixed) and mixing type 4 (MH internally-mixed).

Re: This has been revised as "At 40 and 80 nm sizes, the transition from externally-mixed to MH internally-mixed particles results in a marked increase in $\kappa$, accompanied by a slight rise in σ".

p11, line 228: I am sure that the authors must intend to refer only to the MH internal mixing category and not the LH internal mixing category here.

Re: Revised.

p12, lines 244-26: There does appear to be an increase in the externally-mixed state during the evening rush hour compared to the afternoon just before it. Perhaps you meant the LH internally-mixed state?

Re: The reviewer is right, and we have revised the sentence as "Unlike the 40 nm particles, there is no apparent increase of LH internally-mixed state for the particles with sizes of 80, 110, 150, and 200 nm in the rush hours or cooking times".

p13, line 260: "corresponding period" would seem to refer to the night time which was just discussed, but I assume by "corresponding period" you mean the afternoon, when the LH internally-mixed mode peaks?

Re: The sentence has been revised as "In the early afternoon, the externally-mixed and transitional externally-mixed particles, which represent the intermediate state of the aging process in which particles transition from LH to MH mode, showed a decrease to around 10%, while the proportion

of particles with MH internally-mixed state increased up to 80%".

p13, line 265: changed -> to change

Re: Revised.

p14, line 276: "especially for particles larger than 40 nm" I don't understand why the authors characterize it this way. The difference in the fraction in the MH internally-mixed state is greatest for the 40 nm particles, and I would say that this sentence is not true for the 200 nm particles, where the fraction only achieves a value of ~70%.

Re: The sentence has been revised as "On clear days, the fraction of MH internally-mixed particles increases significantly from ~10-40% before 9:00 to ~70-100% during 12:00–15:00".

p14, line 279: "While" followed by a comma is not an appropriate way to begin a sentence in English. Do the authors mean "However," or "At the same time," or "In addition,"?

Re: The sentence has been revised as "In addition, the fraction of MH internally-mixed particles for 40 nm is noticeably higher on clear days (80%), which is approximately 40% greater than that on cloudy days, indicating that photochemical processes can make particles more internally-mixed and aged (Liu et al., 2021a)". And according to the comment, we have carefully checked and revised the usage of "while" in the manuscript.

p14, lines 284-285: If I interpret "large sizes" to be 150 nm or 200 nm, mean κ decreases from 9:00 through to 17:00 according to Fig. 5.

Re: This sentence has been removed in the revised version.

p20, line 412: the large difference -> there is a large difference

Re: Revised.

Figure S1: Please indicate in the caption the time length of these back-trajectories.

Re: Revised.